# The Na⁺/Ca²⁺, K⁺ exchanger NCKX4 is required for efficient cone-mediated vision

**Frans Vinberg[1], Tian Wang[2,3,4], Alicia De Maria[1], Haiqing Zhao[5], Steven Bassnett[1], Jeannie Chen[2,3,4]\*, Vladimir J Kefalov[1]\***

[1]Department of Ophthalmology and Visual Sciences, Washington University, St. Louis, United States; [2]Zilkha Neurogenetic Institute, Keck School of Medicine, University of Southern California, Los Angeles, United States; [3]Department of Cell and Neurobiology, Keck School of Medicine, University of Southern California, Los Angeles, United States; [4]Department of Ophthalmology, Keck School of Medicine, University of Southern California, Los Angeles, United States; [5]Department of Biology, Johns Hopkins University, Baltimore, United States

**Abstract** Calcium (Ca²⁺) plays an important role in the function and health of neurons. In vertebrate cone photoreceptors, Ca²⁺ controls photoresponse sensitivity, kinetics, and light adaptation. Despite the critical role of Ca²⁺ in supporting the function and survival of cones, the mechanism for its extrusion from cone outer segments is not well understood. Here, we show that the Na⁺/Ca²⁺, K⁺ exchanger NCKX4 is expressed in zebrafish, mouse, and primate cones. Functional analysis of NCKX4-deficient mouse cones revealed that this exchanger is essential for the wide operating range and high temporal resolution of cone-mediated vision. We show that NCKX4 shapes the cone photoresponse together with the cone-specific NCKX2: NCKX4 acts early to limit response amplitude, while NCKX2 acts late to further accelerate response recovery. The regulation of Ca²⁺ by NCKX4 in cones is a novel mechanism that supports their ability to function as daytime photoreceptors and promotes their survival.

**\*For correspondence:** jeannie@ med.usc.edu (JC); Kefalov@vision. wustl.edu (VJK)

**Competing interests:** The authors declare that no competing interests exist.

## Introduction

Local and transient changes in cytosolic Ca²⁺ concentration regulate a wide variety of cellular processes such as synaptic transmission, muscle contraction, and gene expression (*Berridge et al., 2000*). Cytosolic Ca²⁺ concentration reflects a dynamic balance between influx and efflux pathways, operating through plasma membrane channels or across intracellular stores such as endoplasmic reticulum and mitochondria. One mechanism for Ca²⁺ efflux from cells uses the electrochemical gradients of both Na⁺ and K⁺ to extrude Ca²⁺ via the SLC24 family of Na⁺/Ca²⁺, K⁺ exchangers (NCKX1-5). Roles for SLC24 proteins in physiology and disease are beginning to emerge (*Herzog et al., 2015*; *Li and Lytton, 2014*; *Parry et al., 2013*; *Schnetkamp, 2013*) and in several cases appear to involve Ca²⁺ regulation in sensory neurons. One Na⁺/Ca²⁺, K⁺ exchanger with a well-established function is NCKX4 (encoded by *Slc24a4*), which is expressed in olfactory sensory neurons. NCKX4-mediated extrusion of Ca²⁺ from the cilia of olfactory receptor cells shapes the olfactory response and mediates sensory adaptation (*Stephan et al., 2011*). Another family member, NCKX1, is expressed strongly in rod photoreceptors and is the dominant mechanism for extruding Ca²⁺ from their outer segments (*Reiländer et al., 1992*; *Vinberg et al., 2015*). As a result, NCKX1 plays a critical role for maintaining the dynamic equilibrium of Ca²⁺ in rod outer segments and is

**eLife digest** Cells known as photoreceptors sense light in the eye. Light activates signaling pathways inside the photoreceptors that relay visual information to nerve cells, which carry the information to the brain. Photoreceptors called cone cells allow us to distinguish different colors of light and therefore play an important role in daytime vision. Over the course of the day, the overall levels of light in the environment can vary widely and so photoreceptors need to be able to adjust their signaling pathways so that they can still respond to light stimuli.

Calcium ions modulate the signaling pathways inside cone cells to help them adjust to changing light levels. These ions also play other important roles in the health and activity of photoreceptors, so the cells need to carefully control how many calcium ions they contain.

Cone cells contain a structure known as the outer segment, which is responsible for detecting light stimuli. It is believed that cones control the levels of calcium ions in the outer segment by balancing the flow of calcium ions into and out of the segment. The calcium ions enter the outer segment via channels that sit in the membrane surrounding the cell. A transporter protein known as NCKX2, which is only found in cone cells, was thought to pump the calcium ions out of the cell. However, recent data has challenged this idea by demonstrating that NCKX2 only plays a minor role in this process.

Vinberg et al. investigated how calcium ions leave the outer segments of cone cells in several different animals. The experiments show that a transporter protein called NCKX4 – which belongs to the same protein family as NCKX2 – is the main transporter involved in removing calcium ions from the cone cells of mice. Loss of NCKX4 from mouse cones reduced the ability of these cells to respond to fast and rapidly changing light stimuli, and to operate in bright light.

Further experiments show that NCKX4 is also found in the outer segments of zebrafish and monkey cone cells. The next challenges will be to find out if NCKX4 is also present in human cones and whether it plays a role in regulating our daytime vision.

essential for the normal development, function, and survival of rods and for rod-mediated dim light vision (*Reiländer et al., 1992*; *Riazuddin et al., 2010*; *Vinberg et al., 2015*).

The mechanisms that regulate the extrusion of $Ca^{2+}$ from the cone photoreceptors, remain controversial. As this process mediates light adaptation, it is critical for the ability of cones to function as our daytime photoreceptors. The prevailing view is that cones also express a cell-specific $Na^+/Ca^{2+}$, $K^+$ exchanger, NCKX2, that mediates the extrusion of $Ca^{2+}$ from their outer segments (*Prinsen et al., 2000*), a role analogous to that of NCKX4 in olfactory neurons or NCKX1 in rods. However, in vivo electroretinogram recordings from NCKX2-deficient mice failed to detect any functional deficits in their cones (*Li et al., 2006*), raising doubts about the role of NCKX2 in regulating cone $Ca^{2+}$ dynamics. A recent more detailed analysis of isolated cone responses in NCKX2-deficient mice revealed delayed cone response recovery but normal light sensitivity and light adaptation (*Sakurai et al., 2016*). These results suggest the existence of additional, as yet unidentified, mechanism(s) for extruding $Ca^{2+}$ from cone outer segments.

The $Ca^{2+}$ influx/efflux balance in cone photoreceptor outer segments is regulated by the activity of the phototransduction cascade. In the dark, part of the continuous current entering the outer segment through transduction cGMP-gated (CNG) channels is carried by $Ca^{2+}$ (*Miller and Korenbrot, 1987*; *Picones and Korenbrot, 1995*), which is then extruded through mechanisms that remain unclear (*Yau and Nakatani, 1984*). Following photoactivation, the closure of CNG channels blocks $Ca^{2+}$ influx but $Ca^{2+}$ extrusion continues until a new equilibrium is reached. As a result, light activation is accompanied by a decline in the concentration of outer segment $Ca^{2+}$ (*Sampath et al., 1999*; *Yau and Nakatani, 1985*). This triggers the $Ca^{2+}$-mediated negative feedback on phototransduction, a process required for the timely recovery of the light response and for the adaptation of photoreceptors to background light (*Fain et al., 2001*; *Nakatani and Yau, 1988*; *Sakurai et al., 2011*). This strong modulation of cone phototransduction allowed us to use functional analysis of cone photoreceptors to investigate the mechanisms of $Ca^{2+}$ extrusion from their outer segments.

## Results

### The olfactory Ca²⁺ exchanger NCKX4 is expressed in the outer segments of mouse cones

To identify candidate proteins that might contribute to calcium homeostasis in cones, we examined published microarray data from rod-dominant wild type (WT) and cone-dominant NRL-deficient (*Nrl⁻/⁻*) mouse retinas (*Corbo et al., 2007*). We noted that the Na⁺/Ca²⁺, K⁺ exchanger *Nckx4* (*Slc24a4*) was strongly upregulated in the retina of *Nrl⁻/⁻* mice, suggesting that NCKX4 could potentially be present in cone photoreceptors. This observation is also consistent with results of a recent study on differential expression of genes in rods and cones, where NCKX4 and NCKX2 (*Slc24a2*) were identified as cone-specific genes (*Hughes et al., 2017*). To establish the identity of the retinal cells expressing *Nckx4*, we performed in situ hybridization experiments on retinal sections from WT and *Nrl⁻/⁻* mice. We found sparse expression of *Nckx4* in cells at the top of the photoreceptor layer (ONL) of WT mouse retinas (*Figure 1a*). The density and location of the expressing cells suggested that they were cones. Consistent with this notion, the *Nckx4* transcript was abundant in the cone-like photoreceptors of *Nrl⁻/⁻* mice (*Figure 1b*). Together these results demonstrate that *Nckx4* is expressed in cone but not in rod photoreceptors.

To determine the subcellular localization of NCKX4 within cone photoreceptors, we developed a polyclonal NCKX4 antibody (see Materials and methods). Co-localization of peanut agglutinin (PNA), known to label cone outer segments, with the NCKX4 antibody demonstrated that NCKX4 expression was confined mainly to the cone outer segments (*Figure 2a*, top row). Next, we obtained *Nckx4*-floxed mice (*Nckx4^f/f^*, [*Stephan et al., 2011*]) and crossed them with the cone-specific HGRP Cre mouse line (*Cre⁺*, [*Le et al., 2004*]) to generate cone-specific *Nckx4* conditional knock-out (*Nckx4^f/f^ Cre⁺*, see Materials and methods and below) mice. The expression of Cre recombinase alone had no effect on the presence of NCKX4 in cones (*Figure 2a*, middle vs. top rows). However, in *Nckx4^f/f^ Cre⁺* mice, NCKX4 immunofluorescence was absent from the cones (*Figure 2a*, bottom row). In contrast, consistent with the cone-specific expression of Cre, the strong NCKX4 staining in the inner nuclear layer was not diminished in *Nckx4^f/f^ Cre⁺* mice. Comparable results (not shown) were obtained with another NCKX4 antibody, recently shown to specifically recognize NCKX4 (*Bronckers et al., 2017*). Further validation of the NCKX4 antibody by western blot demonstrated that it reacts with a ~50 kDa protein as was observed by Bronckers et al. (*Figure 2b*). Thus, the NKCX4 antibody was selective for NCKX4 and did not cross-react with other proteins expressed in rod or cone photoreceptors. Together, these results indicate that the conditional knockout of NCKX4 in cones was successful. Importantly, the density and morphology of PNA-stained NCKX4-

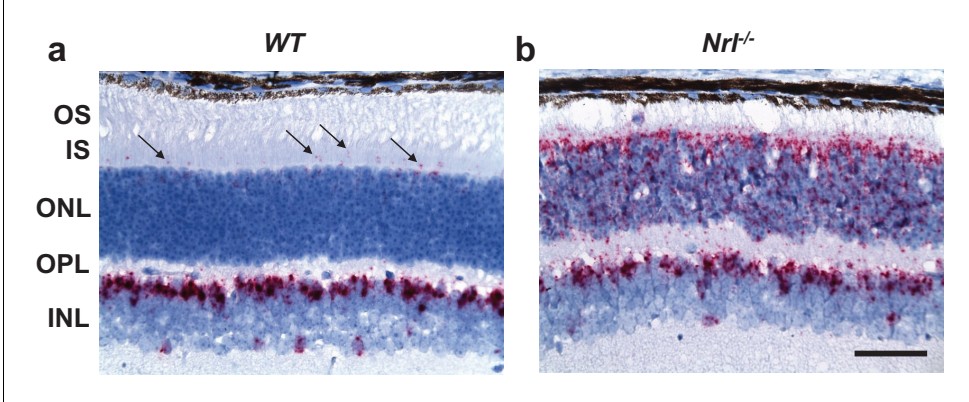

**Figure 1.** NCKX4 is expressed in cone photoreceptors. In situ hybridization with *Nckx4* (*Slc24a4*) probe demonstrating sparse expression of NCKX4 in the photoreceptor layer of a *WT* mouse retina (**a**, arrows) and strong expression of NCKX4 in the photoreceptor layer of an *Nrl⁻/⁻* ('cone-only') mouse retina (**b**). A robust expression of NCKX4 is also evident in the inner nuclear layers of WT and *Nrl⁻/⁻* retinas (**a**, **b**). OS, outer segment; IS, inner segment; ONL, outer nuclear layer; OPL, outer plexiform layer; INL, inner nuclear layer. Scale bar = 50 µm.

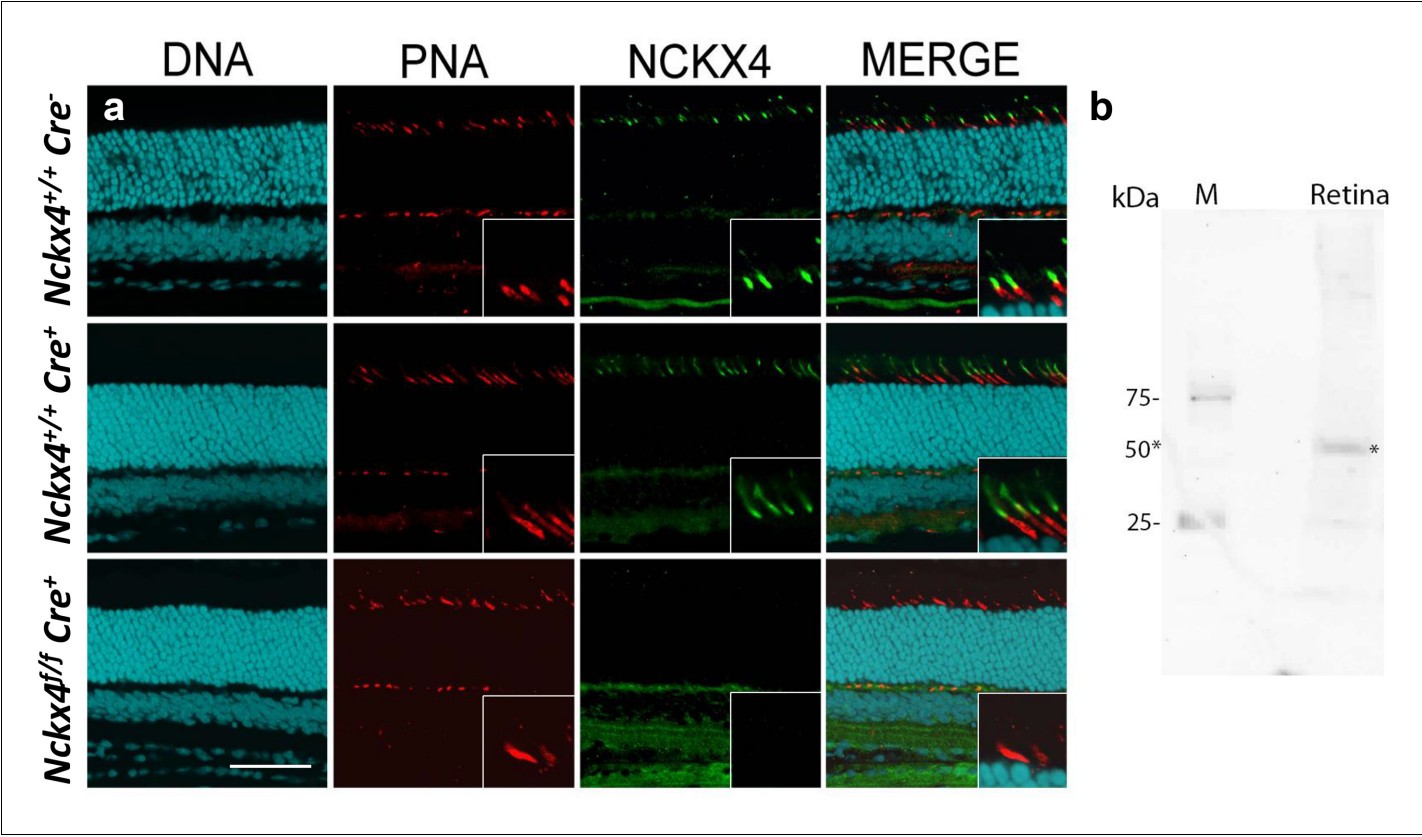

**Figure 2.** NCKX4 is expressed in the outer segments of cone photoreceptors. (a) Immunostaining for NCKX4 in vertical sections of mouse retinas (photoreceptors on the top). Nuclei (DNA, cyan), cone photoreceptors (PNA, red), and NCKX4 (green) staining in Cre-negative $Nckx4^{+/+}$ (top row), Cre-positive control $Nckx4^{+/+}$ $Cre^+$ (middle row) and $Nckx4^{f/f}$ $Cre^+$ (bottom row) mice. Insets show larger magnification immunostaining for cones in the photoreceptor layer. Scale bar = 50 μm. (b) Western blot of wild-type mouse retinal homogenate revealing a protein band of ~50 kDa (*) consistent with NCKX4.

deficient cones were indistinguishable from these of wild type cones, suggesting that the absence of NCKX4 did not affect adversely cone survival.

To further investigate the cone-specific expression of NCKX4, whole mount retina was prepared from control C57BL/6 mice and co-stained for the shortwave cone pigment (S-opsin) and NCKX4. In the dorsal retina, only a subset of NCKX4-positive cells were also co-labeled with S-opsin (*Figure 3a*), a result consistent with the low number of cones that express S-opsin in this region of the retina (*Szél et al., 1992*). In the ventral region, where S-opsin is expressed more uniformly, nearly all cones co-expressed S-opsin and NCKX4 (*Figure 3b*). Thus, NCKX4 is expressed in both M- and S-cones in the mouse retina, indicating that this exchanger could play a role in regulating cone $Ca^{2+}$ and, hence, cone function.

In situ hybridization revealed that NCKX4-positive cells were not restricted to the photoreceptor layer but they were also present in the distal inner nuclear layer (INL, *Figure 1a*). The location of their cell bodies suggested that these NCKX4-postive cells might be rod bipolar cells. To test this hypothesis, we co-labeled retinal sections from control C57BL/6 mice for NCKX4 (*Figure 3c*) and the rod bipolar cell marker PKCα (*Figure 3d*), whereupon strong overlapping immunofluorescence signals were observed (*Figure 3e*). These results suggest that in addition to its presence in the outer segments of mouse cone photoreceptors, NCKX4 might be expressed in rod bipolar cells, where it could potentially be involved in regulating retinal synaptic transmission and signal processing. However, further experiments will be required to confirm the expression of NCKX4 in rod bipolar cells and to determine its potential role for rod signaling.

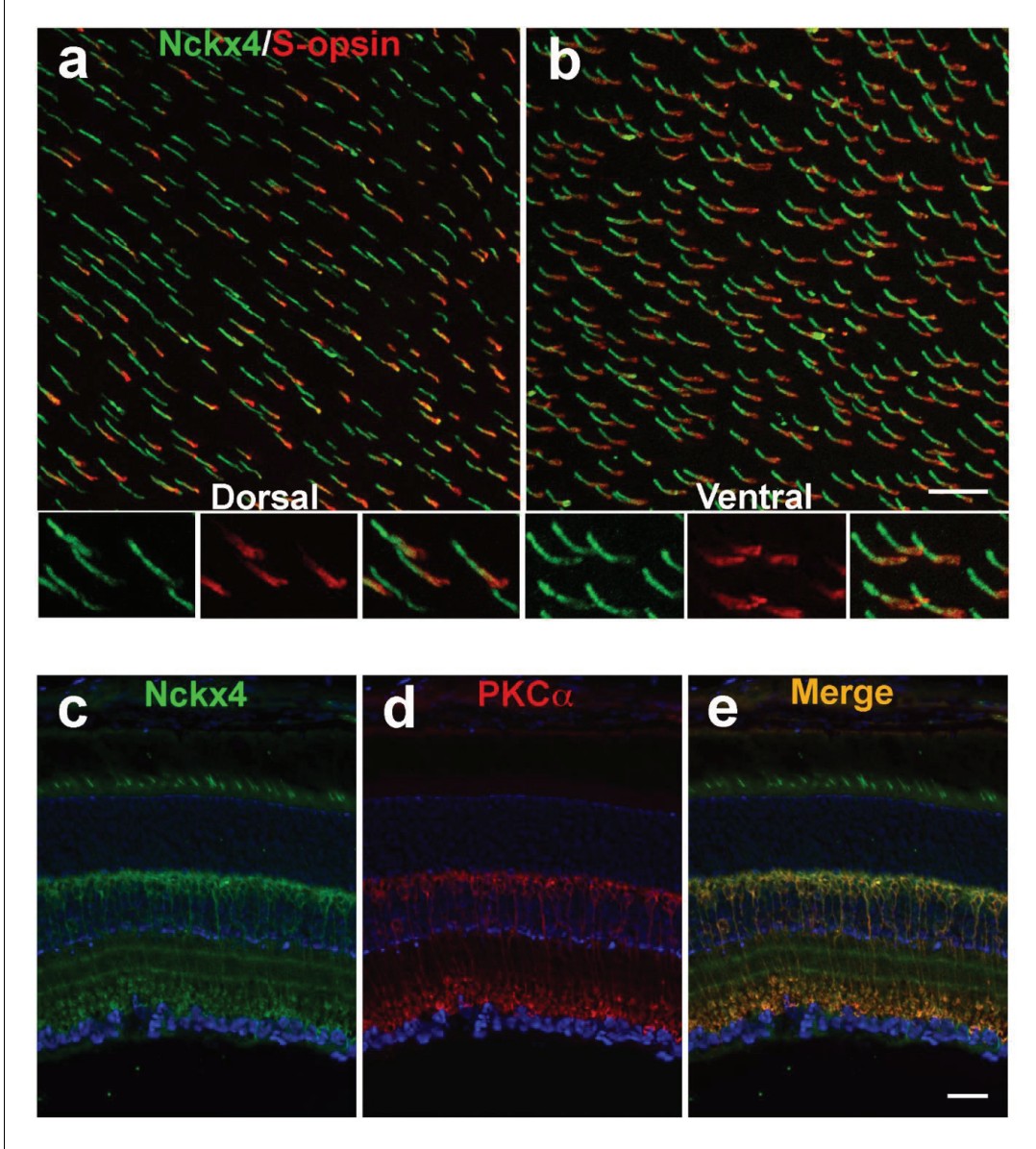

**Figure 3.** NCKX4 is broadly expressed in cones and in rod bipolar cells. Immunostaining for NCKX4 (green) and short-wave opsin (S-opsin, red) in flat mounted retinas. In the dorsal region, S-opsin expressing cones were only a fraction of the NCKX4-expressing cones (a) consistent with the low density of S-cones and high density of M-cones in the dorsal mouse retina. In contrast, nearly all NCKX4-expressing cones in the ventral region expressed S-opsin as well (b) A higher magnification of the labeled cones is shown at the bottom of each panel. S opsin staining appeared stronger in the inner segment and tapered off toward the outer segment, whereas NCKX4 labeling appeared uniform in the outer segment. Immunostaining of retinal sections show NCKX4 expression in cells in the inner nuclear layer (c) Staining of the same tissue section with PKCα (d) a rod bipolar cell marker, shows extensive overlap (e) Scale bars = 20 μm.

## NCKX4 is required for the fast inactivation of mouse cone phototransduction

To determine the role of NCKX4 in cone phototransduction, we first recorded electrical responses from individual dark-adapted mouse cones to flashes of light in $Cre^+$ control and $Nckx4^{f/f} Cre^+$ mouse retinal slices (*Nikonov et al., 2006*) (see *Figure 4a* and Materials and methods). To facilitate the isolation of cone responses, all recordings were done from mice lacking the α-subunit of rod transducin ($Gnat1^{-/-}$) which prevents rods from generating light responses while leaving the structure

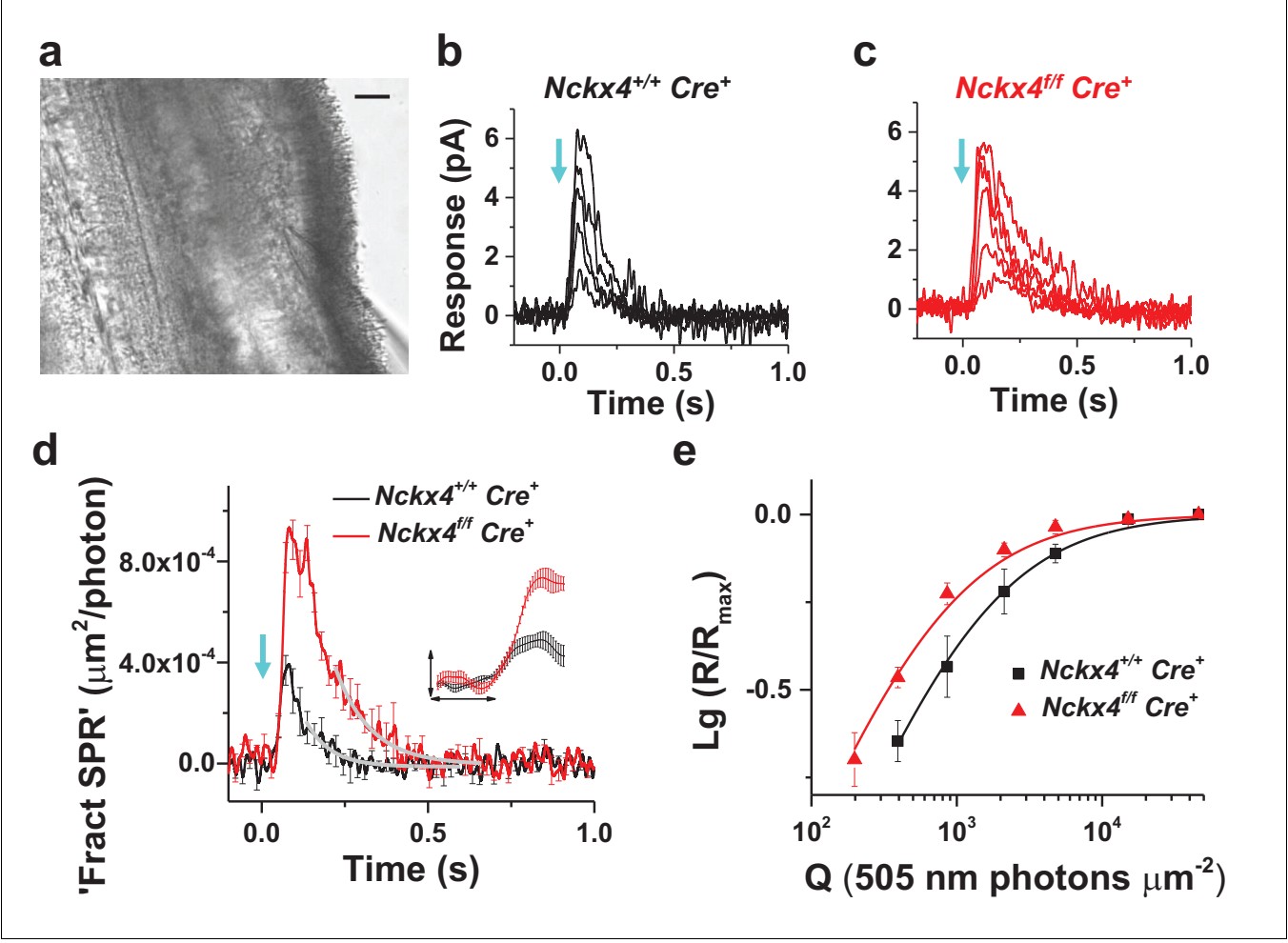

**Figure 4.** NCKX4 accelerates light response termination and decreases the sensitivity of mouse cones. (a) Slice preparation under infrared illumination. Photoreceptor outer segments are pointing to the right and the recording glass pipette is visible at the bottom right corner. Scale bar = 20 μm. Representative light responses recorded from a single inner segment of a control $Nckx4^{+/+}$ $Cre^+$ (b) and $Nckx4^{f/f}$ $Cre^+$ (c) cone. Light flashes (flash length = 1 ms, λ = 505 nm, flash strength Q = 200–46, 100 photons μm$^{-2}$) were delivered at t = 0 s (arrow). (d) Population averaged (mean ± SEM) responses to a dim flash normalized with Q (in photons μm$^{-2}$) and maximal response amplitude ($R_{max}$) recorded from $Nckx4^{+/+}$ $Cre^+$ control (black, Q = 860 photons μm$^{-2}$, N = 6 cells from two mice) and $Nckx4^{f/f}$ $Cre^+$ (red, Q = 393 photons μm$^{-2}$, N = 6 cells from three mice) cones. The tail of the responses is fit by a single exponential function (*Equation 1*) with τ = 75 ms and 106 ms in control and NCKX4-deficient cones, respectively. The inset shows the rising phase of dim flash responses on a finer time scale. The horizontal bar measures 50 ms and the vertical bar is 0.04%. (e) Normalized population averaged response amplitudes ($R/R_{max}$) are plotted as a function of flash strength in photons μm$^{-2}$ for control (black, N = 6 cells from two animals) and NCKX4-deficient cones (red, N = 6 cells from three animals). Smooth traces plot *Equation 3* with $Q_{1/2}$ = 1390 photons μm$^{-2}$ and 730 photons μm$^{-2}$ for control (black) and $Nckx4^{f/f}$ $Cre^+$ (red) cones, respectively.

and function of cones intact (*Calvert et al., 2000*; *Nikonov et al., 2006*). The amplitudes of saturated cone responses ($R_{max}$) to bright test flashes that closed all transduction CNG channels were not affected by the absence of NCKX4 (*Figure 4b,c* and *Table 1*). Thus, the absence of NCKX4 did not cause substantial change in the dark current of cones, suggesting a normal complement of CNG channels and normal [cGMP] in darkness. However, the overall kinetics of the flash responses appeared slower in NCKX4-deficient cones. Comparing responses to identical dim-flash stimuli, we found that response termination was delayed dramatically in the absence of NCKX4 (*Figure 4d*). Quantitative analysis revealed an almost two-fold increase of the time-to-peak ($t_p$) and the recovery time constant of the tail phase of dim flash response ($τ_{rec}$), as well as a two-fold increase in integration time ($t_{int}$) upon deletion of NCKX4 (*Table 1*).

**Table 1.** Comparison of control and NCKX4-deficient cone flash response and light adaptation parameters. $R_{max}$, maximal flash response amplitude (in pA) in single-cell recordings; $t_p$, time from flash to the peak amplitude (in ms) of a dim flash response in single-cell recordings; $\tau_{rec}$, the recovery time constant (in ms) of the tail phase of a dim flash response in single-cell recordings (see **Equation 1**); $Q_{1/2}$, flash strength (in photons $\mu m^{-2}$) eliciting 50% of the $R_{max}$ in single-cell recordings (see **Equation 3**); $\tau_1$ and $\tau_2$, time constants in **Equation 2** describing the recovery kinetics of ex vivo ERG signal after step onset (see **Figure 4c,d**); $I_0$, background light intensity (in photons $\mu m^{-2} s^{-1}$) reducing the flash response sensitivity of cones to 50% of that in darkness as derived from ex vivo ERG data; $T_{int}$, integration time of dim flash responses (defined as the area between baseline and response divided by the peak amplitude) Statistics for parameters (mean ± SEM) derived from ex vivo ERG data ($\tau_1$, $\tau_2$ and $I_0$) were from five control and eight $Nckx4^{f/f} Cre^+$ mice, and the statistics for the other parameters from single-cell recordings were from six control cells (two mice) and six $Nckx4^{f/f} Cre^+$ cells (three mice). * indicates significant (p<0.05) difference between control and NCKX4-deficient cones.

| | $R_{max}$ (pA) | $t_p$ (ms) | $\tau_{rec}$ (ms) | $Q_{1/2}$ (hv $\mu m^{-2}$) | $\tau_1$ (ms) | $\tau_2$ (s) | $I_0$ (hv $\mu m^{-2} s^{-1}$) | $T_{int}$ (ms) |
|---|---|---|---|---|---|---|---|---|
| $Nckx4^{+/+} Cre^+$ | 5.8 ± 1 | 76 ± 3 | 52 ± 8 | 1,150 ± 240 | 109 ± 27 | 1.3 ± 0.2 | 49,000 ± 11,000 | 80 ± 14 |
| $Nckx4^{f/f} Cre^+$ | 5.6 ± 0.4 | 121 ± 6* | 101 ± 13* | 670 ± 73* | 212 ± 37* | 1.5 ± 0.3 | 11,508 ± 990* | 160 ± 9* |

Analysis of the early leading edge of the light response (normalized with $R_{max}$) can be used to assess the onset and efficiency of the phototransduction activation reactions, i.e. the amplification of the light signal by the transduction machinery (**Nikonov et al., 2000**; **Pugh and Lamb, 1993**). The leading edge of the NCKX4-deficient cone response was similar to that of control cones (**Figure 4d**, inset), indicating that the activation of the phototransduction reactions was not affected by the absence of NCKX4. Consistent with this, analysis of the amplification constant in control and NCKX4-deficient cones revealed comparable (p>0.05) values of 0.5 ± 0.1 $s^{-2}$ (n = 6) and 0.8 ± 0.3 $s^{-2}$ (n = 6), respectively. The response onset delay $t_d$ (see **Equation 5**) appeared to increase slightly (p=0.02) in $Nckx4^{f/f} Cre^+$ cones ($t_d$ = 35 ± 2 ms) compared to $Cre^+$ controls ($t_d$ = 29 ± 2 ms). The reason for this subtle change is not evident. Thus, the efficiency of phototransduction activation was not compromised by deletion of NCKX4 but light responses were slower, particularly in their recovery, in the $Nckx4^{f/f} Cre^+$ cones compared to control mice, resulting in larger responses (**Figure 4d**) and increased sensitivity of cones lacking NCKX4 (**Figure 4e**; **Table 1**). Together these results demonstrate that NCKX4 is important for the timely recovery of cone phototransduction. This conclusion is consistent with the hypothesis that NCKX4 participates in the extrusion of $Ca^{2+}$ from cone outer segments, and thus allows a faster decline of $Ca^{2+}$ concentration to accelerate light response termination.

## NCKX4 shifts the operating range of cones to brighter light and accelerates light adaptation

Rapid and efficient adaptation of cones to varying ambient illumination is required for proper visual function. Sluggish sensitivity adjustment in varying background light intensity may, for example, compromise our ability to drive a car in twilight or during the night when large changes in average illumination levels occur rapidly. To study the role of NCKX4 in cone light-adaptation, we measured the sensitivity of $Cre^+$ control and $Nckx4^{f/f} Cre^+$ mouse cones under various background light intensities, using ex vivo ERG recordings (**Vinberg et al., 2014**, **2015**) (see **Figure 5a** and Materials and methods). We began by measuring the kinetics of cone light adaptation in response to near half-saturating steps of light (see legend of **Figure 5** and **Figure 6c**). Both $Cre^+$ control and $Nckx4^{f/f} Cre^+$ cones responded to the onset of a light step with initial hyperpolarization followed by an adaptation-mediated relaxation to a plateau (**Figure 5b**). Thus, despite the absence of NCKX4, cones were able to undergo light adaptation in response to steady background light. However, the response relaxation appeared slower in NCKX4-deficient cones compared to that in control cones, suggesting a delay in $Ca^{2+}$ extrusion. The kinetics of light adaptation in response to a step of light could be fit by a sum of two exponential functions in both control (**Figure 5c**) and $Nckx4^{f/f} Cre^+$ cones (**Figure 5d**). The faster time constant $\tau_1$ was about 100 ms in control cones and increased significantly upon deletion of NCKX4, whereas the slower time constant was ~1.5 s, both in control and $Nckx4^{f/f} Cre^+$ cones (**Table 1**). The proportion of the faster component in the fittings ($A_1/A$, see

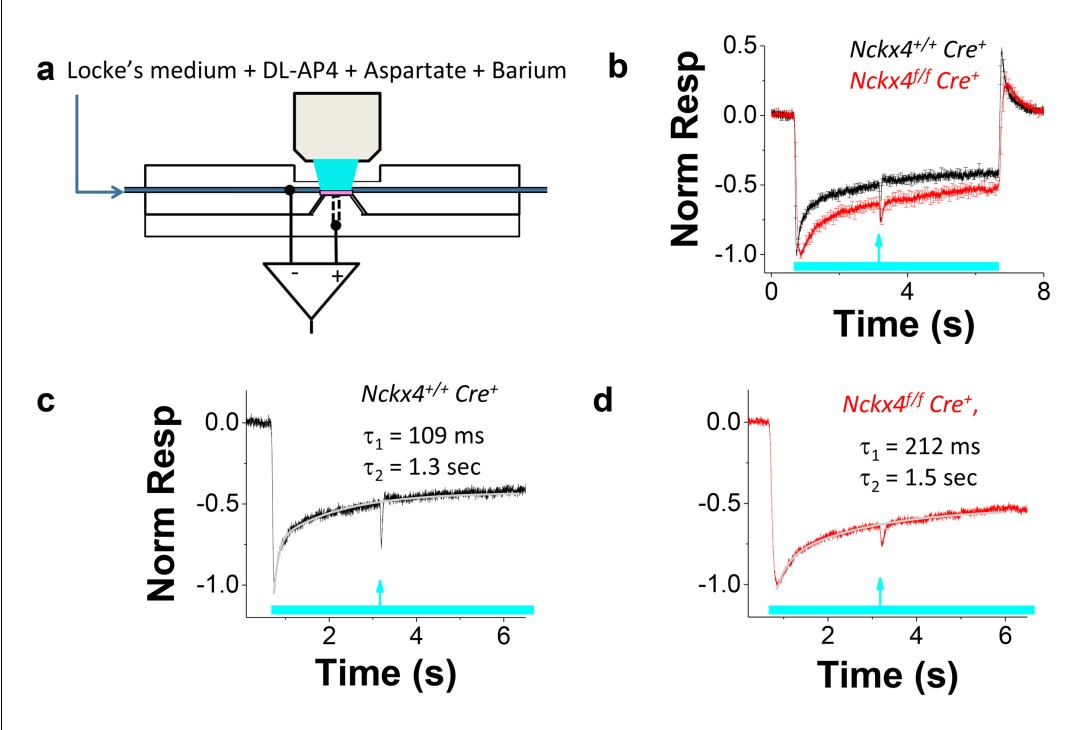

**Figure 5.** NCKX4 accelerates cone light adaptation. (a) A schematic of the system to record transretinal voltage (ex vivo ERG) from a superfused isolated mouse retina (pink, see Materials and methods for details). (b) Population averaged ex vivo cone ERG responses (mean ± SEM) to steps of light normalized to the peak amplitude recorded from $Nckx4^{+/+} Cre^+$ control (black, background light intensity I = 37,800 photons µm$^{-2}$ s$^{-1}$, N = 5 mice) and $Nckx4^{f/f} Cre^+$ (red, I = 15,500 photons µm$^{-2}$ s$^{-1}$, N = 5 mice) retinas. A test flash (arrow) was delivered 2.5 s after the onset of each light step (bar) to probe the sensitivity of cones at different background light intensities (see below). A sum of two exponential functions (*Equation 2*) was fitted to the recovery phase of the averaged step responses (after the step onset) shown in (b) for $Nckx4^{+/+} Cre^+$ control (c) and $Nckx4^{f/f} Cre^+$ (d) mice. The values of the best-fitting time constants are indicated in each panel.

*Equation 2*) was not affected by the absence of NCKX4 ($A_1/A = 46 \pm 10\%$ in control and $A_1/A = 51 \pm 10\%$ in $Nckx4^{f/f} Cre^+$ mice). These results demonstrate that NCKX4 is required for the rapid light adaptation of cones. However, they also indicate the presence of additional, relatively slow and NCKX4-independent, mechanism(s) for extruding $Ca^{2+}$ from cone outer segments.

Next, we probed the sensitivity of $Cre^+$ control (*Figure 6a*) and $Nckx4^{f/f} Cre^+$ (*Figure 6b*) cones to a flash of light (flash response amplitude/flash energy) as it approached steady state, 2.5 s after the onset of background light. As has been described for many sensory neurons, including photoreceptors, the sensitivity of control cones declined as a function of background light according to the Weber-Fechner law (*Figure 6c*). Despite the important role for NCKX4 in setting the temporal properties and dark adapted sensitivity of cones and accelerating light adaptation demonstrated above, at steady state cones lacking NCKX4 still adapted according to the Weber-Fechner law (*Figure 6c*). However, the background light that reduced cone sensitivity two-fold ($I_0$) shifted to about five-fold lower intensity in the absence of NCKX4 (*Figure 6c*, *Table 1*). This shift in the operating range matched well the combined effect of two-fold longer integration time and about 2.3-fold higher sensitivity of dim flash responses of dark adapted $Nckx4^{f/f} Cre^+$ cones as compared to the control cones (see *Table 1* and *Figure 4d*). Thus, removing NCKX4 shifted the operating range of cones to dimmer background lights, effectively restricting the ability of these photoreceptors to function in bright light. This finding demonstrates that NCKX4 is important for setting the operating range of the cones for daytime vision. However, the persistent Weber-Fechner adaptation in steady state light indicates that additional mechanism(s) for extruding $Ca^{2+}$ and sustaining adaptation also exist in these cells.

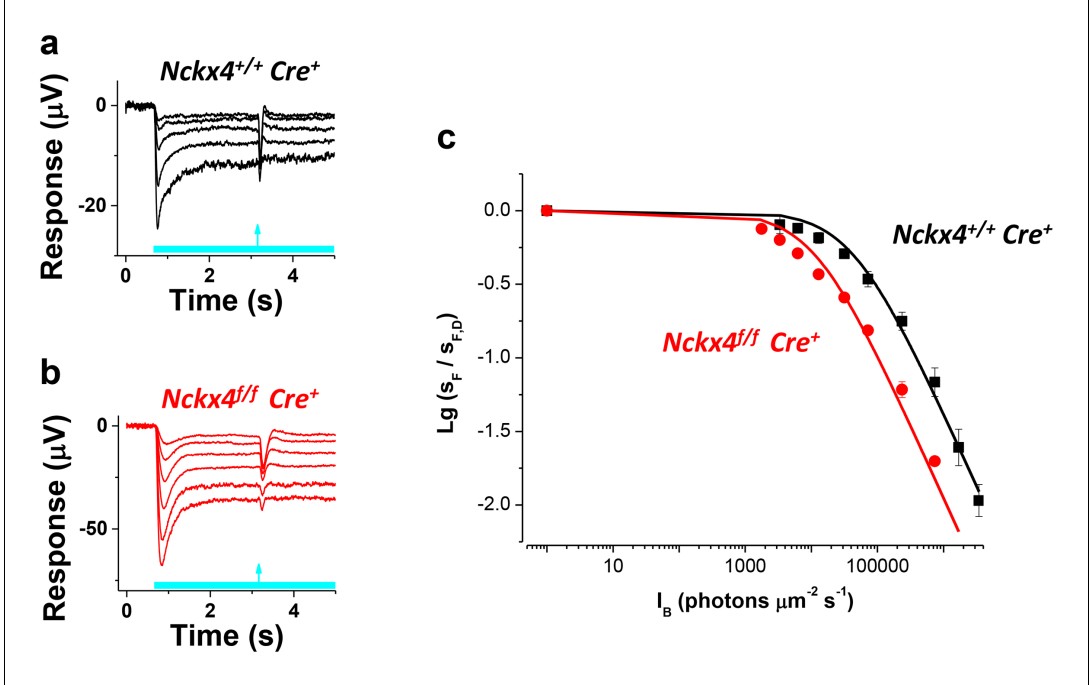

**Figure 6.** NCKX4 extends the function of cones to brighter light. The sensitivity to a flash of light ($S_F$) for $Nckx4^{+/f}$ $Cre^+$ control (a) and $Nckx4^{f/f}$ $Cre^+$ (b) cones in background light was determined 2.5 s after the light step onset. Light step and flash timing are indicated on the bottom of each panel. (c) $S_F$ as normalized to the sensitivity in darkness ($S_{F,D}$, mean ± SEM) is plotted as a function of light step intensity ($I_B$) for $Nckx4^{+/+}$ $Cre^+$ control (black, N = 5 retinas) and $Nckx4^{f/f}$ $Cre^+$ (red, N = 5 retinas) mice. Smooth traces plot the Weber-Fechner function (**Equation 4**) with $I_0$ = 43,000 and 11,300 photons $\mu m^{-2}$ $s^{-1}$ for control (black) and NCKX4-deficient (black) cones, respectively.

## NCKX4 is required for the high temporal resolution of cone-mediated vision

As our data above demonstrates, NCKX4 is important for the timely recovery of the light response in cones. Thus, we also sought to determine if this delay in NCKX4-deficient cones compromises the temporal resolution of cone-mediated vision. We assessed the temporal frequency range of the cone retinal signaling pathway in vivo by recording ERG responses to flickering light from anesthetized $Cre^+$ control and $Nckx4^{f/f}$ $Cre^+$ mice. At low frequencies, the cone b-wave (positive ERG wave dominated by cone-driven bipolar cell activity in the retina [*Shirato et al., 2008*]) of both control and NCKX4-deficient mice could faithfully follow the flicker stimulus (see 5 Hz flicker stimulation in *Figure 7a and b*). However, consistent with our single-cell recordings, the b-wave responses from $Nckx4^{f/f}$ $Cre^+$ mice recovered more slowly than these from $Cre^+$ control mice. The responses in control mice could easily follow the stimulus for frequencies up to 20 Hz (*Figure 7a*). In contrast, as a result of their slower termination, the responses of NCKX4-deficient mice began to overlap and saturate at frequencies as low as 10 Hz and could not follow the flicker stimulus at higher frequencies (*Figure 7b*). Consistent with this observation, analysis of the fundamental response amplitude (R, measured from negative to positive peak) revealed significantly steeper decline with increasing flicker frequency in $Nckx4^{f/f}$ $Cre^+$ mice compared to $Cre^+$ controls (*Figure 7c*). These results demonstrate that NCKX4 plays an important role in enhancing the temporal resolution of cone-mediated signaling in the retina.

## Cones require both NCKX2 and NCKX4 for normal structure and function

Cones singly deficient in either NCKX2 (*Sakurai et al., 2016*) or NCKX4 (this study) still exhibited normal steady-state adaptation and by inference, persistent $Ca^{2+}$ extrusion from the outer segment. Thus, we generated mice lacking both exchangers in their cones in order to examine their functional

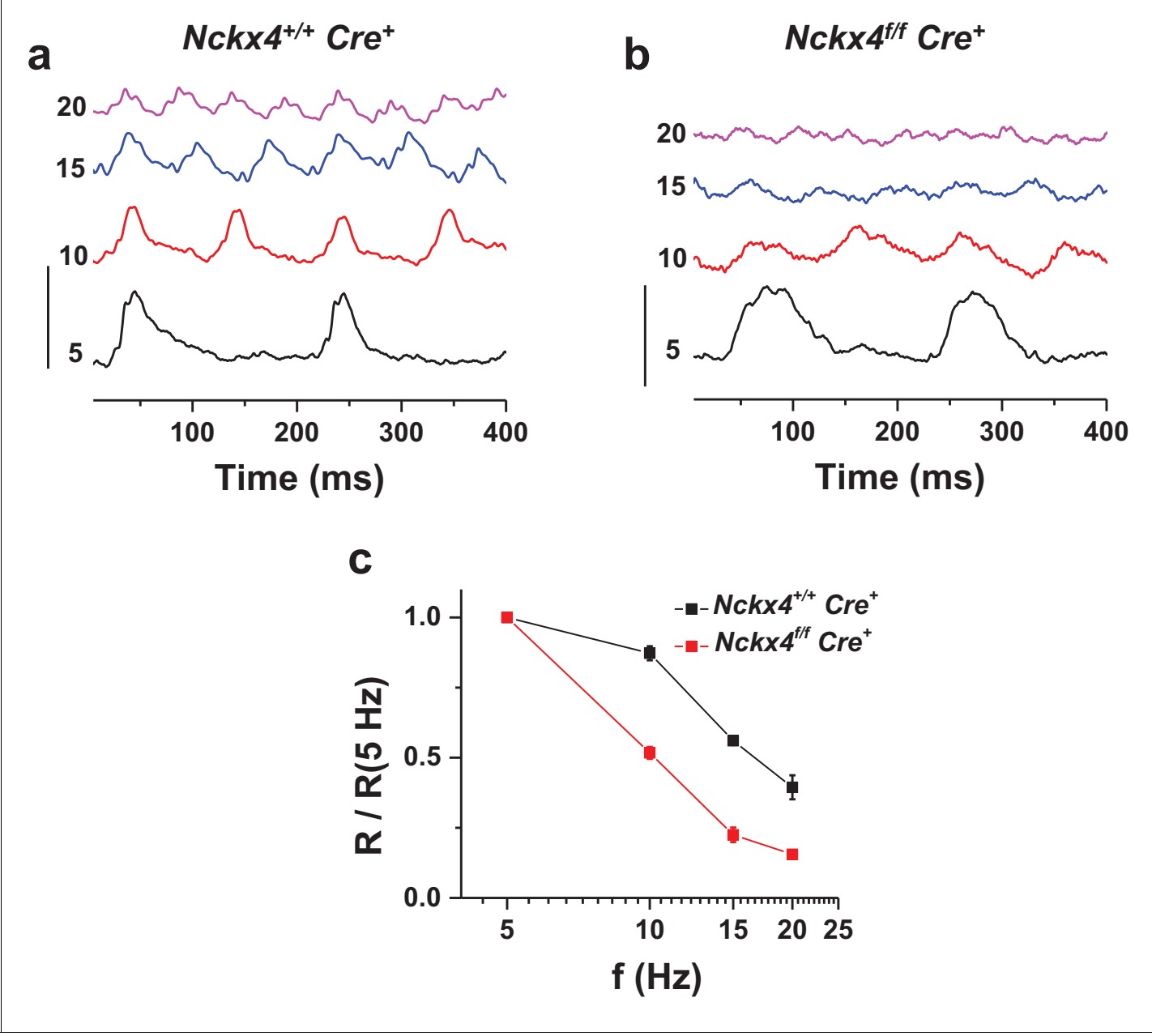

**Figure 7.** NCKX4 extends the temporal resolution of cone-mediated vision to higher frequencies. In vivo electroretinogram (ERG) responses to flickering light recorded from dark-adapted anesthetized $Nckx4^{+/+} Cre^+$ (a) and $Nckx4^{f/f} Cre^+$ (b) mice. The frequency of the flicker stimulus (f) is indicated on the left for each trace in Hz. Vertical scale bar = 3 μV. The energy of flickering flashes were either 0 or 0.5 lg(Cd m$^{-2}$ s). (c) The fundamental response amplitude (R), measured from the most negative peak/plateau to the most positive peak and normalized to the fundamental response amplitude at 5 Hz stimulation, is plotted as a function of the flicker stimulus for $Nckx4^{+/+} Cre^+$ (black, N = 5 eyes from three animals) and $Nckx4^{f/f} Cre^+$ (red, N = 8 eyes from four animals) mice.

redundancy and determine whether additional mechanisms of Ca$^{2+}$ extrusion exist in the cone outer segment. First, we prepared retinal flat mounts to investigate the effect of the simultaneous removal of these two exchangers on cone number and outer segment morphology. The absence of NCKX2 alone has no discernible effect on these parameters, as we have previously shown (*Sakurai et al., 2016*). Cone outer segments from the ventral region of $Nckx2^{-/-}$ retinas from 3-month-old mice labeled with S-opsin (*Figure 8a*) or the α-subunit of the cone cGMP-gated channel (CNGA3;

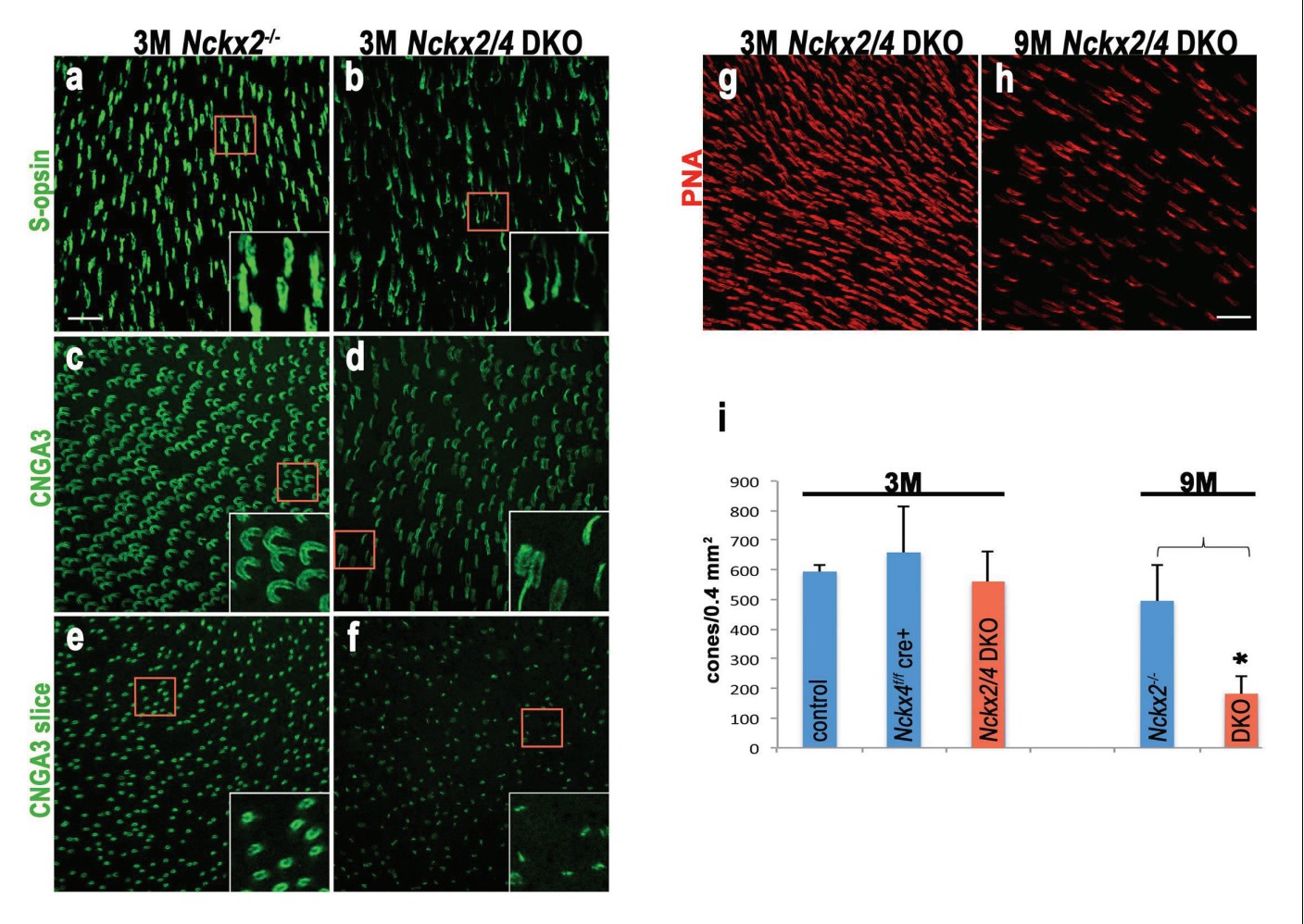

**Figure 8.** Loss of both NCKX2 and NCKX4 expressions leads to alteration of outer segment structure and cone cell death. Confocal stacked images from flat mounted retinas from 3-month-old *Nckx2*[-/-] mice (**a, c, e**) and their littermate *Nckx2/4* double knockout animals (**b, d, f**) stained with antibodies against S-opsin (**a, b**) or CNGA3 (**c, d**). A slice from the CNG3A stacked image is shown in e and f. Higher magnification of representative cells are shown in inset. Cones were labeled with peanut agglutinin (PNA) in representative flat mounted retina from 3-month-old (**g**) and 9-month-old (**h**) *Nckx2/4* double knockout mice. Scale bar = 20 µm. The number of cones from retinas of control, *Nckx4* knockout, and *Nckx2/4* double knockout mice were counted and plotted (**i**). One-way ANOVA showed no differences in cone numbers in the group of 3-month-old mice (p=0.3), but significant (p<0.0001) reduction in cone numbers in the 9-month-old *Nckx2/4* double knockouts compared to *Nckx2*[-/-] mice.

*Figure 8c*) antibodies appeared homogeneous in diameter and morphology. A tangential slice from a CNGA3-labeled flat mount showed largely circular cross-sections of cone outer segments, consistent with the localization of the CNG-gated channel at the plasma membrane (*Figure 8e*). In contrast, the outer segment diameter of cones from mice lacking both exchangers varied, as seen in S-opsin (*Figure 8b*) and CNGA3 (*Figure 8d*) labeled cells. A cross-section of the CNGA3-labeled flat mount showed numerous cellular profiles of lines instead of circles, as if the circular structures had collapsed (*Figure 8f*). Despite this change in outer segment structure, the number of PNA-positive cones was not statistically different between *Gnat1*[-/-] control, *Nckx4* conditional knockout, and *Nckx2/4* double knockout (*Nckx2/4 DKO*) cones in 3-month-old mice (*Figure 8i*). At 9 months of age, however, strikingly fewer number of PNA-positive cones were detected in the *Nckx2/4* double knockout cones when compared to Cre-negative *Nckx2*[-/-]*Nckx4*[f/f] controls (*Figure 8g–i*, p<0.0001, two-tailed *t*-test). Thus, the simultaneous deletion of *Nckx2* and *Nckx4* resulted in abnormal cone outer segment morphology and progressive cone death.

We next investigated how the ablation of NCKX2 and NCKX4 expression affects the function of cones. Consistent with our observation from single-cell suction recordings (*Figure 4*), comparison of flash response families obtained with transretinal recordings demonstrated that the response recovery is delayed greatly in NCKX4-deficient cones compared to controls (*Figure 9*, compare a and b). The maximal response amplitudes were similar, consistent with the observed normal cone numbers and cone morphology in NCKX4-deficient cones (*Figures 2*, *8*). However, the simultaneous deletion of the two exchangers resulted in a dramatic reduction in the maximal response of NCKX2/4 double knockout (DKO) cones (*Figure 9c*). The delay in cone response recovery caused by the absence of NCKX4 (*Figure 9b*) or NCKX2 (*Figure 9d*, inset) was extended even further when both exchangers were removed simultaneously (*Figure 9d*), demonstrating the complementary roles of NCKX2 and NCKX4 in mediating the extrusion of Ca$^{2+}$ from cone outer segments. *Nckx2/4 DKO* cone sensitivity was also significantly reduced compared to both control and NCKX4-deficient cones (*Figure 9e*). These results are consistent with the observed morphologic alteration of outer segment structure in these cones (*Figure 8*) and demonstrate that cone function was severely affected by the simultaneous ablation of NCKX2 and NCKX4. Together, the abnormal morphology and severe functional

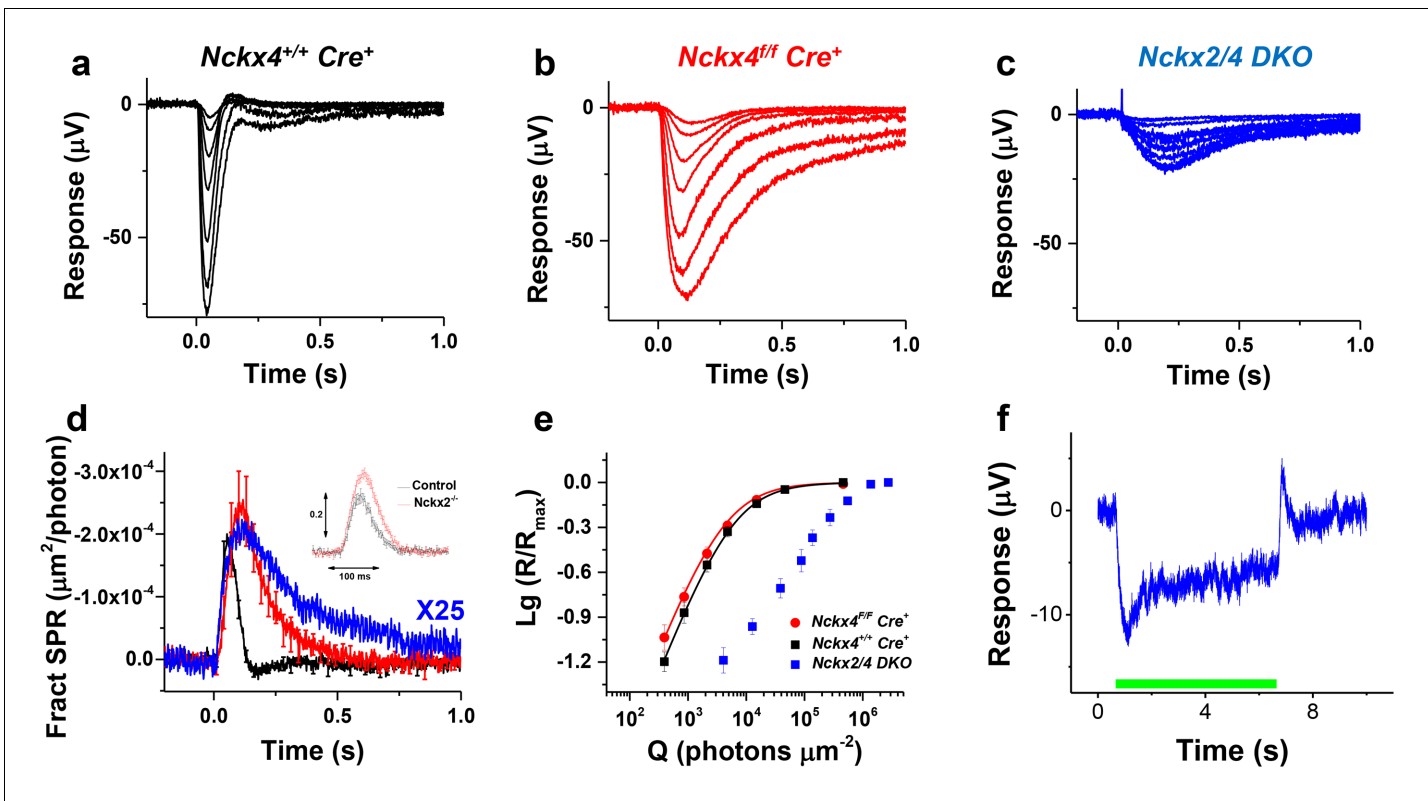

**Figure 9.** Loss of both NCKX2 and NCKX4 expressions severely compromises cone function. Representative light responses recorded from isolated retinas of *Nckx4$^{+/+}$ Cre$^+$* (a), *Nckx4$^{f/f}$ Cre$^+$* (b) and *Nckx2$^{-/-}$ Nckx4$^{f/f}$ Cre$^+$* (c) mice using ex vivo ERG method. Light flashes ranged from 390 to 460,000 photons (505 nm) µm$^{-2}$ in (a) and (b), and from 4000 to 1.4 * 10$^6$ photons (505 nm) µm$^{-2}$. (d) Population averaged (mean ± SEM) responses to a dim flash normalized with Q (in photons µm$^{-2}$) and maximal response amplitude (R$_{max}$) recorded from *Nckx4$^{+/+}$ Cre$^+$* control (black, N = 5 retinas from four mice), *Nckx4$^{f/f}$ Cre$^+$* (red, N = 8 retinas from four mice) and *Nckx2$^{-/-}$ Nckx4$^{f/f}$ Cre$^+$* (blue, N = 4 retinas from three mice) retinas. The response from *Nckx2$^{-/-}$ Nckx4$^{f/f}$ Cre$^+$* mice shown in (d) is scaled up by 25-fold to facilitate comparison of response kinetics between the genotypes. The inset shows fractional dim flash responses of control (WT) and *Nckx2$^{-/-}$* cones modified from (*Sakurai et al., 2016*). (e) Normalized population averaged response amplitudes (R/Rmax) are plotted as a function of flash strength in photons µm$^{-2}$ for *Nckx4$^{+/+}$ Cre$^+$* control (black, N = 5), *Nckx4$^{f/f}$ Cre$^+$* (red, N = 8), and *Nckx2$^{-/-}$ Nckx4$^{f/f}$ Cre$^+$* (blue, N = 4) mouse retinas. Smooth traces plot *Eq. 3* with Q$_{1/2}$ = 5200 photons µm$^{-2}$ and 3900 photons µm$^{-2}$ for control (black) and *Nckx4$^{f/f}$ Cre$^+$* (red) retinas, respectively. (f) Light adaptation persists in the cones from *Nckx2$^{-/-}$ Nckx4$^{f/f}$ Cre$^+$* mice. Representative response to a bright step of 530 nm light (I = 38,600,000 photons µm$^{-2}$ s$^{-1}$) recorded from *Nckx2$^{-/-}$ Nckx4$^{f/f}$ Cre$^+$* mouse retina. Light step timing is indicated on the bottom of the graph.

deficits in cones lacking both exchangers suggest that the combined activity of NCKX2 and NCKX4 represents the dominant mechanism by which calcium is extruded from cone outer segments.

Finally, we also examined the ability of NCKX2/NCKX4-deficient cone photoreceptors to adapt to background light. The small residual signaling and reduced light sensitivity of *Nckx2/4 DKO* cones resulted in substantial pigment bleaching by the step of light, which made the analysis challenging and prevented us from quantifying the results. However, the response to a step of light revealed substantial relaxation 1–2 s after the onset of the background light (*Figure 9f*). This surprising result suggests that these cones might still be able to undergo light adaptation despite the block of both NCKX2- and NCKX4-driven extrusion of $Ca^{2+}$. Further studies will be needed to determine whether this adaptation is mediated by a slow NCKX-independent extrusion of $Ca^{2+}$ from cone outer segments or by a $Ca^{2+}$-independent mechanisms that become unmasked upon the deletion of NCKX2 and NCKX4.

## Expression of NCKX4 in zebrafish and macaque cone outer segments

Our molecular and functional analyses demonstrated that NCKX4 is expressed in mouse cones where it plays an important role in regulating their function. To investigate whether the use of NCKX4 by cones is a widespread phenomenon conserved throughout evolution, we examined the retinas of zebrafish and macaque. PNA staining was used to identify the abundant cone photoreceptors in the zebrafish retina (*Figure 10a*). NCKX4 immunofluorescence was observed in several cell types in the zebrafish retina, consistent with our observation on the mouse retina. In the outer retina, however, the NCKX4 signal was restricted to cones (*Figure 10a*, inset), demonstrating that similar to the case of mouse, zebrafish cone photoreceptors also express NCKX4.

Three spectrally distinct cone photoreceptors mediate high-acuity daytime color vision in humans. To explore whether NCKX4 could potentially also play a role in human daytime vision, we analyzed its expression in macaques, primates with retinas that are very similar to the human retina. Our NCKX4 antibody immunolabeled the cone outer segments (identified by PNA labeling) in sections of

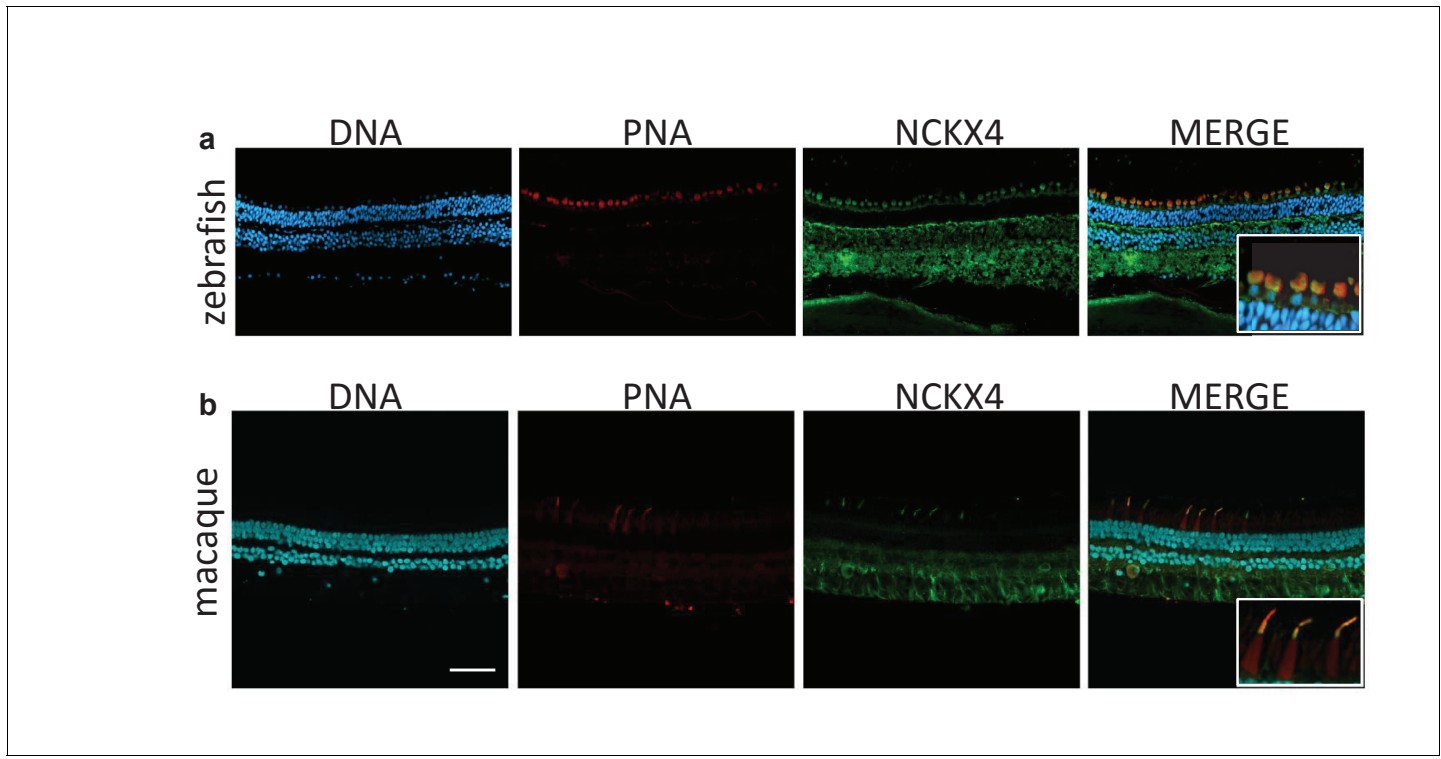

**Figure 10.** NCKX4 is expressed in the outer segments of zebrafish and non-human primate cones. Immunostaining of NCKX4 in the vertical sections of zebrafish (top) and macaque (bottom) retinas (photoreceptors on the top). Nuclei (DNA, cyan), cone photoreceptors (PNA, red), and NCKX4 (green) staining. Insets show larger magnification immunostaining of cones in the photoreceptor layer. Scale bar = 50 µm.

the mid-peripheral region of the macaque retina (*Figure 10b*), demonstrating the expression of NCKX4 in primate cones. Together, these results indicate that the expression of NCKX4 in cones is conserved from zebrafish to mice and non-human primates, suggesting that NCKX4 could potentially also play a role in cone-driven primate and human daytime vision.

## Discussion

Novel physiological functions have begun to emerge for NCKX proteins encoded by the *SLC24* gene family. In addition to previously suggested roles for NCKX1 in rods and night vision/blindness (*Neuillé et al., 2016*; *Reiländer et al., 1992*; *Riazuddin et al., 2010*; *Vinberg et al., 2015*), NCKX2 in cones (*Prinsen et al., 2000*; *Sakurai et al., 2016*) and brain (*Li et al., 2006*; *Tsoi et al., 1998*), and NCKX4/5 in hair/eye/skin pigmentation (*Lamason et al., 2005*; *Sulem et al., 2007*), recent studies have demonstrated a role for NCKX4 in olfaction (*Stephan et al., 2011*), satiety (with nonsense mutation leading to anorexia in mice) (*Li and Lytton, 2014*) and tooth amelogenesis (*Bronckers et al., 2015*; *Herzog et al., 2015*; *Hu et al., 2012*; *Parry et al., 2013*; *Seymen et al., 2014*; *Wang et al., 2014*). Here we report for the first time that NCKX4 is expressed in fish, mouse, and non-human primate cone (but not rod) photoreceptor outer segments and is required for rapid cone response recovery, light adaptation, and normal photopic vision. Our results provide explanation for the surprisingly mild cone phenotype of $Nckx2^{-/-}$ mice (*Li et al., 2006*; *Sakurai et al., 2016*) by demonstrating that NCKX4 is critical for the phototransduction feedback driven by extrusion of $Ca^{2+}$ from cone outer segments. We propose that use of two exchangers in cones may contribute to their unique functional properties, including rapid photoresponses, efficient adaptation, and a dynamic range suitable for daytime function. Such duplicity could also protect against severe visual disorders caused by mutations in individual *Nckx2* or *Nckx4* genes.

### Role of NCKX4 in cone phototransduction and light adaptation

The main effect of NCKX4 removal is to slow the cone light response recovery and to increase the peak amplitude of subsaturating flash responses (*Figure 4*), indicating slower $Ca^{2+}$ extrusion and phototransduction feedback. Detailed comparison of dim flash response waveforms from control and NCKX4-deficient cones demonstrates that the efficiency of phototransduction activation reactions is not affected by the presumptive slower $Ca^{2+}$ extrusion (*Figure 4d*). Interestingly, dark-adapted NCKX4-deficient cone flash responses resemble those of GCAPs knockout cones in which the $Ca^{2+}$ feedback that accelerates cGMP synthesis is absent (*Sakurai et al., 2011*). Hence, NCKX4 appears to be important for rapidly lowering $[Ca^{2+}]$ in cones following light-induced closure of CNG channels to mediate the $Ca^{2+}$ feedbacks in the time scale of flash responses (~500 ms).

$Ca^{2+}$ contributes significantly to both rod and cone light adaptation (*Fain et al., 2001*; *Nakatani and Yau, 1988*). Without $Ca^{2+}$ feedback, it has been suggested that rod cells would simply integrate single photon responses and saturate at very dim background light intensities (*Matthews et al., 1988*; *Nakatani and Yau, 1988*). Thus, disrupting $Ca^{2+}$ extrusion is expected to compromise light adaptation. Indeed, we found that deletion of NCKX4 delays light adaptation and shifts the operating range of cones to dimmer light (*Figures 5* and *6*). However, cones without NCKX4 still adapt according to the Weber-Fechner law, indicating that $Ca^{2+}$ is still extruded from their outer segments. The lack of detectable cone degeneration (*Figures 2* and *8*) and the normal cone light response amplitudes of NCKX4-deficient cones (*Figures 4* and *9*) also support the existence of additional $Ca^{2+}$ extrusion pathways in cones. Our results from cones lacking both NCKX2 and NCKX4 (*Figure 9*) suggest that the $Ca^{2+}$ extrusion from NCKX4-deficient cones is mainly via NCKX2. This exchanger is expressed in chicken, human and mouse cones, where it has been shown to regulate cone phototransduction (*Prinsen et al., 2000*; *Sakurai et al., 2016*). We propose that the combined activity of NCKX2 and NCKX4 is required for the rapid and efficient extrusion of $Ca^{2+}$ from cones and that this enables these cells to adapt rapidly and remain functional in a wide range of light backgrounds throughout the day.

Interestingly, adaptation was apparent even in cones lacking both NCKX2 and NCKX4 (*Figure 9f*). Our results do not allow us to distinguish between calcium-dependent and calcium-independent mechanisms of light adaptation in NCKX2/NCKX4-deficient cones. One possibility is that residual $Ca^{2+}$ extrusion somehow persists even in the combined absence of NCKX2 and NCKX4. Indeed, slow NCKX-independent extrusion of $Ca^{2+}$ has been recently suggested for rod outer

segments (*Vinberg et al., 2015*). Such a mechanism might be common to all photoreceptors, potentially serving as a safety valve for $Ca^{2+}$ release. Alternatively, it is possible that some residual calcium-independent adaptation exists in cones and becomes unmasked when $Ca^{2+}$ extrusion is blocked in NCKX-deficient cones.

By comparing the functional deficiency of cones lacking NCKX4 with the recent results from cones lacking NCKX2 (*Sakurai et al., 2016*), it is possible to identify the primary mechanism for $Ca^{2+}$ extrusion in cones. Whereas the deletion of both NCKX4 and NCKX2 causes a notable delay in the cone response recovery, the effect is substantially more pronounced in NCKX4-deficient cones than in NCKX2-deficient cones (*Figure 9d*). In addition, the amplitude of the single photon response and, consequently, cone sensitivity are also substantially (>2 fold) increased in the absence of NCKX4 compared to controls (*Figure 4*), whereas a similar analysis of NCKX2-deficient cones revealed only subtle increase in the single photon response amplitude and no change in cone sensitivity (*Sakurai et al., 2016*). Thus, the extrusion of $Ca^{2+}$ that modulates phototransduction after photoactivation appears to be dominated initially by NCKX4, with NCKX2 contributing mostly during the late phase of the flash response (*Figure 9d*). Such temporal separation could be the result of different kinetics or ionic equilibrium of inactivation of the two exchangers, enabling them to differentially modulate $Ca^{2+}$ extrusion. Regardless of the molecular mechanisms that regulate the activities of NCKX2 and NCKX4, our findings lead to the conclusion that NCKX4 is the dominant $Ca^{2+}$ extrusion pathway in cone outer segments.

## Redundant roles of NCKX2 and NCKX4 in cone survival

Whereas the number of cones was similar between single knockout of NCKX2, NCKX4 and control $Gnat1^{-/-}$ mouse retinas, removal of both exchangers had a marked effect on cone structure/function and survival. Results in the DKO retinas are similar to the findings from NCKX1-deficient rods, where the removal of the sole exchanger presumably renders the cells unable to efficiently extrude $Ca^{2+}$. Importantly, lowering of $Ca^{2+}$ stimulates retinal guanylyl cyclase to synthesize cGMP through the action of GCAPs (*Dizhoor et al., 1995*; *Gorczyca et al., 1994*; *Mendez et al., 2001*; *Palczewski et al., 1994*). In support of this mechanism, we measured lowered cGMP levels in NCKX1-deficient retinas (*Vinberg et al., 2015*). Due to the high cooperativity of the cGMP-gated channel, lowered cGMP provides an explanation for the large decrease in light sensitivity in $Nckx1^{-/-}$ retinas. We hypothesize that loss of both NCKX2 and NCKX4 in cones has the same effect as loss of NCKX1 in rods. That is, in the absence of NCKX-driven $Ca^{2+}$ extrusion, cGMP synthesis is not stimulated, and more cGMP-gated channels remain closed, leading to the observed dramatic decrease in response amplitude and light sensitivity (*Figure 9*). Over the long-term, the disrupted $Ca^{2+}$ homeostasis may gradually lead to cone death.

## Differences between rods and cones

The striking differences in adaptation capacity and kinetics of daytime color vision and dim-light vision can be attributed in part to the distinctive physiological properties of rods and cones, that is cones have faster light response kinetics and are able to remain functional under brighter light than rods. However, the molecular origins of these differences remain incompletely understood (*Ingram et al., 2016*). Biochemical evidence from non-mammalian species have shown faster decay of Meta II (active form of visual pigment) (*Shichida et al., 1994*), higher expression of RGS9 and resulting faster inactivation of PDE (*Tachibanaki et al., 2012*), faster synthesis rate of cGMP (*Takemoto et al., 2009*) and higher arrestin content (*Tomizuka et al., 2015*) in cones compared to rods. Such differences could potentially explain the faster function of cones compared to rods. However, direct physiological evidence of the significance of these molecular differences is still lacking, and it is not known if these differences hold in mammalian photoreceptors. As light adaptation in both rods and cones is mediated by the light-induced decline in $Ca^{2+}$ (*Matthews et al., 1988*; *Nakatani and Yau, 1988*), it is reasonable to assume that the better light adaptation capacity of cones is mediated by more efficient $Ca^{2+}$ extrusion and $Ca^{2+}$ feedback. However, recent results indicate that one of the main $Ca^{2+}$ feedbacks, mediated by GCAP1/2, contributes similarly to the light adaptation capacity of rods and cones (*Sakurai et al., 2011*). Moreover, although recoverin appears to modulate cone phototransduction somewhat more efficiently than rod phototransduction, its role in light adaptation is quite marginal (*Makino et al., 2004*; *Sakurai et al., 2015*). Thus, the remaining

possibility is that Ca$^{2+}$-dependent modulation of cone CNG channel (*Korenbrot et al., 2013*; *Rebrik et al., 2012*) together with larger and faster changes of Ca$^{2+}$ concentration in the outer segments of cones between dark and bright light conditions (*Sampath et al., 1999*, *1998*) contribute to the faster and wider range of daytime cone-mediated vision compared to rod-mediated nighttime vision. Our data demonstrates that the cone-specific Ca$^{2+}$ exchanger NCKX4 accelerates the responses of cones (*Figure 4*) and cone-mediated vision (*Figure 7*) and shifts the operating range of cones to brighter background lights (*Figure 6*). By doing so, NCKX4 contributes to the unique properties of cones and their ability to operate at brighter light as compared to rods.

## Cell-specificity of NCKX expression in the mouse retina

Our recent work and the results presented here demonstrate that NCKX1 is the sole Na$^+$/Ca$^{2+}$ exchanger expressed in rods, whereas cones express both NCKX2 and NCKX4 (but not NCKX1; [*Sakurai et al., 2016*; *Vinberg et al., 2015*]). We also found evidence for expression of NCKX4 in inner retina neurons identified tentatively as rod bipolar cells (*Figures 1* and *3*) but the physiological relevance of its presence there remains to be addressed. The reason for tissue/cell-specific expression patterns of different NCKX isoforms is still unclear, particularly considering the similarities in their biophysical properties (*Jalloul et al., 2016*). However, it is known that NCKX1 dimers form hetero-oligomers with the rod transduction CNG channel (*Bauer and Drechsler, 1992*; *Poetsch et al., 2001*; *Schwarzer et al., 1997*, *Schwarzer et al., 2000*). Consistent with this notion, our recent results suggest that NCKX1 is important for normal rod channel function and that deletion of NCKX1 leads to reduced expression and lower conductance of rod CNG channels in mice (*Vinberg et al., 2015*). On the other hand, deletion of NCKX2 (*Sakurai et al., 2016*) or of NCKX4 (*Figure 4*) does not affect the dark current of cones. The simplest explanation for this result is that, in contrast to the case in rods, the expression of NCKX does not control the expression or conductance of cone CNG channels. Such a scenario is consistent with biochemical evidence that NCKX2 does not interact directly with CNGA3 in cone outer segments (*Matveev et al., 2008*). However, coexpression of NCKX2 and CNGA3 in HEK293 cells was found to result in direct interaction between these two proteins (*Kang et al., 2003*), indicating that this issue will require further investigation.

Our data here also explains the striking phenotypic difference between deletion of NCKX1 (and nonsense mutation in the *SLC24A1* gene encoding NCKX1 that cause night blindness in humans) leading to 100-fold reduction of rod response amplitudes and severely compromised dim light vision (*Vinberg et al., 2015*), as compared to deletion of either NCKX2 or NCKX4 in cones leading only to altered kinetics of light responses and light adaptation without affecting cone dark current. The redundancy of NCKX2 and NCKX4 may further explain why mutations in NCKX2 or NCKX4 have not been linked to eye diseases. Notably, even cones lacking both NCKX2 and NCKX4 respond robustly to light (*Figure 9*), further emphasizing the difference between rods and cones.

Interestingly, SNPs in *SLC24A4* correlate with lighter eye and hair pigmentation in Europeans (*Sulem et al., 2007*). It is possible that the redundancy in cone exchangers has allowed evolution of the gene coding NCKX4 in Europeans to adapt to environment with lower ambient light levels without at the same time compromising color vision. It may be interesting to test whether Europeans with SNPs in *SLC24A4* (i.e. people with blue/green eyes and blond hair) or those individuals with Amelogenesis Imperfecta, caused by a nonsense *SLC24A4* mutation (*Herzog et al., 2015*), have subtle abnormalities in cone vision, such as altered ERG flicker fusion frequency, confirming the functional importance of NCKX4 in humans.

## Materials and methods

### Animals

We crossed *Nckx4$^{f/f}$* mice in which exon 5 of the NCKX4 encoding gene (*Slc24a4*) is floxed (*Stephan et al., 2011*), with the HRGP cone Cre-expressing (*Le et al., 2004*) *Gnat1$^{-/-}$* (*Calvert et al., 2000*) line that has been backcrossed to C57BL/6J background for several generations, to produce cone-specific *Nckx4$^{f/f}$ Gnat1$^{-/-}$ Cre$^+$* knock-out and *Nckx4$^{+/+}$ Gnat1$^{-/-}$ Cre$^+$* control mice. A set of experiments were also conducted to compare the cone phenotype between *Nckx4$^{f/f}$ Gnat1$^{-/-}$ Cre$^+$* and *Nckx4$^{f/f}$ Gnat1$^{-/-}$ Cre$^-$* as well as *Nckx4$^{+/+}$ Gnat1$^{-/-}$ Cre$^+$* and *Nckx4$^{+/+}$ Gnat1$^{-/-}$ Cre$^-$* littermates.

Unless otherwise stated, mice used for the experiments were 2- to 3-month-old males and females. Genotyping for the floxed/WT *Slc24a4* as well as for the KO/WT *Gnat1* alleles and for HRGP Cre was performed by Transnetyx (Transnetyx, TN) and the mice were also tested to be free of the *Rd8* mutation (*Mattapallil et al., 2012*). Mice were kept under 12/12 hr light/dark cycle with access to water and standard rodent chow. Before the experiments, mice were dark-adapted overnight and euthanized by $CO_2$ asphyxiation. For single-cell and ex vivo electrophysiology experiments, the eyes were removed quickly after euthanasia and retinas were dissected under an infrared light-equipped microscope as described previously (*Vinberg and Kefalov, 2015*). All experimental protocols were in accordance with the Guide for the Care and Use of Laboratory Animals and were approved by the institutional Animal Studies Committee at Washington University.

## In situ hybridization

Mouse enucleated eye globes were fixed for 24 hr in 4% paraformaldehyde/PBS, embedded in paraffin, and sectioned in the midsagittal plane at 4 µm. In situ hybridization was carried out using the RNAscope technique (RNAscope 2.0; Advanced Cell Diagnostics, CA), as per manufacturer instructions. The target probe set was generated against *Slc24a4* transcripts. The probe set consisted of 20 pairs of oligonucleotides spanning a ~1 kb contiguous region of the target mRNA transcript (*Slc24a4* NM_172152.2). As a negative control, some sections were hybridized with target probe against *DapB*, a bacterial gene encoding dihydrodipicolinate reductase, a key enzyme in lysine synthesis. A target probe directed against ubiquitously expressed *PolR2A* (DNA-directed RNA polymerase II polypeptide A) served as a positive control. Following proprietary preamplification and amplification steps, target probes were detected using an alkaline-phosphatase-conjugated label probe with Fast Red as substrate and Gill's Hematoxylin as a counterstain. With this technique, individual mRNA molecules were detected as bright red puncta.

## Immunodetection

Fluorescence immunocytochemistry was performed on paraformaldehyde-fixed (PAF 4% in buffer phosphate saline pH 7.4, 15 min at room temperature) frozen or vibratome sections of zebrafish, mouse, and monkey retina, respectively, or flat mount mouse retina. Blocking and antibody incubation were carried out in 2% bovine serum albumin, 2% goat serum and 0.1% Triton X-100 in buffer phosphate saline (pH 7.4). The antibody against mouse NCKX4 was produced by Primm Biotech, Inc. MA (RRID: AB_10792951) using a recombinant His-Tag Slc24a4 (from amino acid 229 to 366) as antigen. Secondary antibody used was goat anti-rat Alexa 488 (Life Technologies, CA). In all experiments, no-primary antibody control was run in parallel. Similar cone staining was observed with another NCKX4 antibody (N414/25, NeuroMab, UC Davis/NIH NeuroMab Facility) recently shown to specifically recognize NCKX4 (*Bronckers et al., 2017*). Cone photoreceptor outer segments were labeled with peanut agglutinin (PNA) conjugated with Alexa 568 (Life Technologies, CA). Methyl Green (*Prieto et al., 2014*) (emission 663–686 nm) was used as a DNA counterstain. Confocal microscopy was performed on an Olympus Fluoview 1000 microscope or Zeiss LSM5 Pascal microscope.

Samples for western blot analysis were prepared as follows: retinas from two month old mice were homogenized in PBS containing protease inhibitor cocktail (88666, Thermo Scientific Pierce, IL) and the intact cell nuclei were eliminated by centrifugation. Protein content was determined by BCA assay (Bio-Rad, CA) using immunoglobulin G as standard. Proteins (10 µg) and Molecular Weight Standards (Precision Plus Protein Standards, Bio-Rad) were separated on SDS-PAGE (4%–15% Mini-Protean TGX gels, Bio-Rad) and then transferred to nitrocellulose membranes. Blots were incubated overnight at 4°C with rat anti-NCKX4 diluted 1:500. Primary antibody was detected using horseradish peroxidase-conjugated secondary antibodies (Thermo Fisher Scientific, IL).

## Single cell, ex vivo and in vivo ERG electrophysiology

Electrical responses of cone photoreceptors to light stimulation were recorded from single cells using a suction electrode method (*Baylor et al., 1979*; *Nikonov et al., 2006*) and from isolated retinas by ex vivo ERG method (*Arden and Ernst, 1970*; *Winkler, 1972*). To separate the light responses of cones from those generated by rods, all the mice were bred to $Gnat1^{-/-}$ background

rendering rod photoreceptors unresponsive to light without affecting the cone physiology (*Calvert et al., 2000*).

Suction electrode recordings were performed as described previously (*Wang and Kefalov, 2010*) with small modifications. Briefly, the dorsal half (dominated by M-cones) of the retina was flat-mounted photoreceptor-side upwards on a filter paper (HABG01300, Millipore, MA). A manual slicer was then used to cut a 200–250 µm vertical slice that was mounted on the bottom of a specimen holder by attaching the filter paper side of the slice on a wall made from vacuum grease (111, Dow Corning, MI). The slice was perfused at 3 ml/min with 37°C Locke's solution containing (in mM): NaCl, 112; KCl, 3.6; MgCl$_2$, 2.4; 1.2, CaCl$_2$; HEPES, 10; NaHCO$_3$, 20; Na$_2$-succinate, 3; Na-gluta-mate, 0.5; glucose, 10. In addition, the solution was supplemented with 0.1% of MEM vitamins and amino acids (Sigma-Aldrich, MO), and equilibrated with 95%O$_2$/5%CO$_2$ at 37°C. The slice was visual-ized under infrared light with inverted microscope and the proximal outer nuclear layer was targeted with a glass pipette (tip resistance ~0.6 MΩ, see *Figure 4A*) containing (mM): NaCl, 140; KCl, 3.6; MgCl$_2$, 2.4; CaCl$_2$, 1.2; HEPES, 3; and glucose, 10; pH adjusted to 7.4 with NaOH. Small current sig-nals from the cone inner segments were amplified by using a standard Axopatch 200B amplifier, low-pass filtered at 50 Hz (8-pole Bessel, model 3382, Krohn-Hite, MA) and collected at 10 kHz (1440A Digidata, Molecular Devices, CA). A 505 nm LED source (SR-01-E0070, Luxeon Star LEDs, Ontario, Canada) provided calibrated light stimulation as described in detail previously (*Vinberg et al., 2015*).

Ex vivo ERG recordings were performed as described previously (*Vinberg et al., 2015*). Isolated dark-adapted mouse retinas were mounted on the specimen holder (*Vinberg et al., 2014*) photore-ceptor side upwards and perfused 2 ml/min. at 37°C with Locke's medium similar to that used in suc-tion electrode recordings (see above, [*Vinberg and Kefalov, 2015*; *Vinberg et al., 2014*]). The medium was supplemented with 2 mM L-Aspartate, 40 µM DL-AP4 (Tocris Bioscience, UK) and 100 µM BaCl$_2$ to isolate the photoreceptor component of the ERG signal. Transretinal voltage between the electrodes of the specimen holder was amplified (100X) by a differential amplifier (DP-311, Warner Instruments, CT) and data were low-pass filtered and collected at 300 Hz and 10 kHz, respectively, by using the same digitizer and analog low-pass filter as in suction electrode recordings (see above). The same custom-build LED light stimulation was used as in the suction electrode recordings.

Flicker in vivo ERG recordings were performed by using UTAS Visual Diagnostic System with Big-Shot$^{TM}$ Ganzfeld stimulation unit modified for mouse ERG experiments (LKC Technologies, MD). Mice were anesthetized by intraperitoneal injection of ketamine (80 mg/kg) and xylazine (15 mg/kg) cocktail and the eyes were dilated with drops of 1% atropine sulfate. During the recordings the mouse body temperature was maintained at ~37°C by using ATC1000 (World Precision Instruments, FL) temperature control system with mouse heating pad (model 502195, World Precision Instru-ments, FL). DC ERG signal between corneal electrodes (STelesSR, LKC Technologies, MD) and a ref-erence electrode inserted under the skin at the skull as a response to flickering light at frequencies between 5 and 20 Hz were acquired at 1000 Hz with low-pass filter set at 300 Hz. The flash energy of flickering flash was set to either 0 or 5 dB (i.e. 0 or 5 lg(Cd m$^{-2}$ s)).

A single exponential function was used to describe the time course of the tail phase of dim flash responses:

$$r(t) = Ae^{-\frac{t-t_d}{\tau}}, \tag{1}$$

where $(t_d, A)$ is the starting point of the fitted function and $\tau$ is the time constant of the recovery tail, was fitted (free parameter = $\tau$) to the data recorded from single cones (see *Figure 4d*).

A sum of two exponential functions was used to describe the recovery of cone responses follow-ing the onset of light steps:

$$r(t) = r_0 + A_1\left(1 - e^{-\frac{t-t_d}{\tau_1}}\right) + (A - A_1)\left(1 - e^{-\frac{t-t_d}{\tau_2}}\right), \tag{2}$$

where A is the amplitude measured from the starting point $(t_d, r_0)$ to the plateau when $t \to \infty$ and $A_1$ is the fraction of the recovery contributed by the mechanism underlying the faster exponential time constant $\tau_1$. A best fitting function with a set of freely changing parameters $\tau_1$, $\tau_2$ and $A_1$ was fitted to the ex vivo ERG data (see *Figure 5c,d*).

A Naka-Rushton function:

$$\frac{r}{r_{\max}} = \frac{Q}{Q + Q} \tag{3}$$

was fitted to response amplitude (R) data (*Figure 4e*) to determine the flash strength producing 50% of the maximal response amplitude $r_{\max}$ ($Q_{1/2}$).

The Weber-Fechner function:

$$\frac{s_F}{s_{F,D}} = \frac{I_0}{I_0 + I} \tag{4}$$

was fitted to the light adaptation data (see *Figure 6*) to determine the background light intensity (I) at which the sensitivity of cones is 50% of that in darkness ($I_0$). The sensitivity in darkness ($S_{F,D}$) and during background lights ($S_F$) was determined as the response amplitude to a flash producing <0.2 $r_{max}$ divided by the flash strength in photons $\mu m^{-2}$.

The kinetics of phototransduction activation reactions were quantified by using a model originally developed by Lamb and Pugh (*Lamb and Pugh, 1992*)

$$\frac{r}{r_{max}} = 1 - exp\left[-0.5QA(t - t_d)^2\right] \tag{5}$$

where $t_d$ is a small delay and A is the amplification in $s^{-2}$. Both $t_d$ and A were selected to provide the best fit to the early rising phase of a dim flash response (Q = 393–859 photons $\mu m^{-2}$). All the assumptions of the model may not be valid for fast cone responses (see [*Smith and Lamb, 1997*]) but the model was used here only to compare quantitatively the activation kinetics of phototransduction between control and NCKX4-deficient cones. The same model has been used previously by (*Nikonov et al., 2006*) to describe M- and S-cone phototransduction activation.

## Data and statistical analysis

Electrophysiology data analysis, including statistical analysis, fitting and figure preparation, was performed with Origin 9 (OriginLab, MA). Immunohistochemistry images were prepared with Photoshop 11.

All data, where applicable, are presented as mean ± SEM, and the number of mice, retinas and/or cells used in each experiment are indicated in the Figure and Table legends. The statistical significance of difference between parameters calculated from the mutant and control samples was tested with a two-tailed unpaired Student's t-test or with one-way ANOVA (p<0.05).

## Acknowledgements

This work was supported by National Institutes of Health grants, EY027387 (VJK and JC), EY019312 and EY025696 (VJK), EY012155 (JC), EY026651 (FV), EY024607 (SB), DC007395 (HZ), and EY02687 (Washington University, Department Ophthalmology), Research to Prevent Blindness, and the Ella and Georg Ehrnrooth Foundation (FV). We thank Joseph Corbo for his initial insight into the possible expression of NCKX4 in cones. We also thank Jonathan Lytton from the University of Calgary for the *Nckx2*<sup>-/-</sup> mice and Janis Lem from Tufts University for the *Gnat1*<sup>-/-</sup> animals.

## Additional information

### Funding

| Funder | Grant reference number | Author |
|---|---|---|
| Research to Prevent Blindness | | Frans Vinberg<br>Alicia De Maria<br>Steven Bassnett<br>Vladimir J Kefalov |
| Ella ja Georg Ehrnroothin Säätiö | | Frans Vinberg |

| National Eye Institute | EY026651 | Frans Vinberg |
|---|---|---|
| National Institute on Deafness and Other Communication Disorders | DC007395 | Haiqing Zhao |
| National Eye Institute | EY024607 | Steven Bassnett |
| National Eye Institute | EY027387 | Jeannie Chen Vladimir J Kefalov |
| National Eye Institute | EY012155 | Jeannie Chen |
| National Eye Institute | EY019312 | Vladimir J Kefalov |
| National Eye Institute | EY025696 | Vladimir J Kefalov |

The funders had no role in study design, data collection and interpretation, or the decision to submit the work for publication.

## Author contributions

FV, Conceptualization, Data curation, Formal analysis, Writing—original draft, Writing—review and editing; TW, ADM, Data curation, Formal analysis, Writing—review and editing; HZ, Resources, Writing—review and editing; SB, Resources, Supervision, Writing—review and editing; JC, VJK, Conceptualization, Data curation, Formal analysis, Supervision, Funding acquisition, Writing—original draft, Writing—review and editing

## Author ORCIDs

Vladimir J Kefalov, http://orcid.org/0000-0002-1659-008X

## Ethics

Animal experimentation: This study was performed in strict accordance with the recommendations in the Guide for the Care and Use of Laboratory Animals of the National Institutes of Health. All of the animals were handled according to approved institutional animal care and use committee (IACUC) protocols (#A-3381-01) of the University of Washington in St. Louis.

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
