## [Decision Letter]

Thank you for submitting your article "The olfactory Na^+^/Ca^2+^, K^+^ exchanger NCKX4 is required for efficient cone-mediated vision" for consideration by *eLife*. Your article has been favorably evaluated by Richard Aldrich (Senior Editor) and three reviewers, one of whom, Fred Rieke, is a member of our Board of Reviewing Editors. The following individual involved in review of your submission has agreed to reveal his identity: Clint Makino (Reviewer #3).

The reviewers have discussed the reviews with one another and the Reviewing Editor has drafted this decision to help you prepare a revised submission.

All three reviewers felt the paper made an important contribution to our understanding of cone function. Several points emerged as particularly important in the discussion among the reviewers:

1) Antibody validation. The immunohistochemistry is based on a newly developed antibody, and thus it is particularly important to validate its specificity. There are specific suggestions in the reviews, and the Journal of Comparative Neurology standards for reporting immunohistochemistry results also provide good guidelines. It is particularly important to include western blots of WT, NCKX KO and double KO.

2) Rising phase analysis. The lack of effect of NCKX4 deletion on the rising phase of the light response is a key point of interpretation. This needs to be shown more clearly (i.e. on an expanded time base) and quantified.

3) Writing. More care is needed to distinguish the direct observations in the paper (e.g. impact of NCKX deletion on light responses) from inferences (e.g. impact on calcium levels). This is particularly an issue in the discussion.

*Reviewer #1:*

This paper explores the role of NCKX4 in cone photoreceptor function. The main conclusion is that NCKX2 and NCKX4 together regulate cone calcium levels and aspect of phototransduction that depend on calcium. The experiments generally appear well done and the data is clear. I have some concerns about the directness of the conclusions. Most importantly, in my view the paper could be considerably strengthened if a few of the interesting issues raised were explored in more depth.

1) Amplification phase of light response.

An important result in the paper is that NCKX4 deletion does not alter the rising phase of the light response. This is based in Figure 4. But the time base in the figure is too long to compare rising phases with any confidence, and quantification is needed to support the statement that the rising phases are unchanged as stated in the text.

2) Remaining adaptation.

Adaptation appears to persist in the absence of both NCKX2 and NCKX4. This is presented as a surprise, but then is not pursued. The conclusion reached in the paper is that cones possess another means of regulating calcium; this rests on the assumption that there are not calcium-independent forms of adaptation, and I think this assumption should be stated clearly.

3) Lack of NCKX2 data.

The paper is weakened by the fact that the NCKX2 data was published elsewhere, and is not part of the present paper. One of the most interesting aspects of the paper is the apparent functional redundancy of the two proteins. The paper would be much stronger if complementary experiments were presented for cones lacking each protein individually and both together. Could some of the data from the previous paper be replotted here for comparison?

4) Lack of direct measures of calcium.

In much of the paper, changes in response are equated with changes in calcium. In a few instances, the fact that this is an inference and not direct correspondence is clear, but in many cases the implications for changes in calcium are presented too strongly. I think it is quite important to separate the experimental observations from the inferences made about calcium signaling.

*Reviewer #2:*

The manuscript by Vinberg et al. describes carefully executed experiments to address the question of Ca extrusions mechanisms in retinal cone outer segments. It was long assumed that the NCKX2 Na/Ca-K exchanger was the main extrusion mechanism in this preparation based on the presence of abundant NCKX2 transcripts in chicken and human retinal cone photoreceptors and I gather the assumption that no redundancy in the form of other Na/Ca(-K) exchangers was present (as is the case in retinal rod photoreceptors). The fact that the NCKX2-/- mice showed only modest deficiencies in cone responses (as detailed in a previous publication by these authors) clearly showed the above assumptions were incorrect. In this paper, the authors identify NCKX4 as the "additional" Ca extrusion mechanism in mouse cone outer segments and using cone-specific knockdown of NCKX4. The authors combine immunohistochemistry with various methods of analyzing in detail cone photoresponses in retinas of various mouse models to arrive at their conclusions. I believe this is an excellent paper addressing a key outstanding question about the role of Na/ca-K exchangers and I recommend to publish this paper in *eLife* after the authors address the following minor concerns:

1) The immunohistochemistry is all based a novel NCKX4 antibody developed by the authors. I think for new antibodies it would be very helpful to carry out validation by demonstrating reactivity and specificity against in this case NCKX4 expressed in cell lines transfected with NCKX4 cDNA. The authors here show immunohistochemistry of WT and NCKX4 KO animals but only with cone-specific knockdown. Significant immunoreactivity is observed elsewhere in the retina. Have the authors carried out characterization of their antibody in cell lines expressing transfected NCKX4 or have they carried out immunohistochemistry in the NCKX2-4 DKO? Western blots of retinal extracts from WT and the various KO animals would address this well.

2) I noticed that in Figure 2 the PNA and NCKX4 do not really co-localize but are found in different parts of the cone outer segment? This is in contrast to the zebrafish and macaque retinas where co-localization appears to be observed. Can the authors comment on this discrepancy?

3) In the subsection “Cell-specificity of NCKX expression in the mouse retina”, the authors argue that lack of reduced dark current in either the NCKX2 or NCKX4 KO would be consistent with a lack of interaction between CNGA3 and either NCKX2 or NCKX4. Not sure if I am convinced about this argument as it would require careful measurements of NCKX2 and NCKX4 expression and although Matveev et al. failed to detect an interaction between NCKX2 and CNGA3, co-expression of both in HEK293 cells was shown to result in a direct interaction between these two proteins (Kang et al., Biochemistry 42: 4593).

4) I would prefer to see the olfactory deleted from the title. Since NCKX4 was cloned it has been clear that NCKX4 had a broader expression pattern compared with say NCKX1 and the authors list in the introduction a number of other tissues in which NCKX4 has been shown to play an important role.

*Reviewer #3:*

Calcium is a key regulator of phototransduction in rods and cones. Calcium enters the cone outer segment through ion channels and is removed by an exchanger, thought to be NCKX2. In mutant mice that develop an all cone retina at the expense of rods, expression of the olfactory exchanger NCKX4 was upregulated suggesting that it might be part of the cone phototransduction machinery. The present study confirms the hypothesis by mapping its expression in retina to cones and rod bipolar cells and by recording the electrical responses of cones from mice in which either NCKX4 or NCKX2 and NCKX4 expression was blocked. Absence of NCKX4 alone slowed flash response recovery so that the photon response continued to build for a longer time to a larger amplitude and it subsequently recovered slowly. The cones remained fully capable of adapting to light, but their Weber Fechner relation was shifted to lower intensities. Knockout of NCKX2 and 4 caused a slow degeneration but before cones disappeared, they did respond to light. Maximal response amplitude was reduced, flash responses had even slower kinetics and sensitivity was lower by an order of magnitude. The work greatly furthers our understanding of cone physiology and helps to explain the unexpectedly mild phenotype of NCKX2 knockout in cones. It reveals the existence of an additional mechanism for controlling intracellular calcium and raises other interesting issues about the role of calcium that will need to be characterized in future studies. The work should therefore be of general interest to the readership of *eLife*.

[Editors' note: further revisions were requested prior to acceptance, as described below.]

Thank you for submitting your article "The Na^+^/Ca^2+^, K^+^ exchanger NCKX4 is required for efficient cone-mediated vision" for consideration by *eLife*. Your article has been reviewed by two peer reviewers, and the evaluation has been overseen by a Reviewing Editor and Richard Aldrich as the Senior Editor. The following individuals involved in review of your submission have agreed to reveal their identity: Paul Schnetkamp (Reviewer #2).

The reviewers have discussed the reviews with one another and the Reviewing Editor has drafted this decision to help you prepare a revised submission.

The paper is much improved. There is a request for western blots to help validate the new NCKX antibody used in the paper. Including such data – specifically testing whether the antibody recognizes a protein with the correct MW – would help both validate data in the current paper and help determine how useful the new NCKX4 antibody is more generally.

*Reviewer #2:*

I believe the authors have done an excellent job to address this reviewer's comments, and, as far as I can tell, also the comments by the other two reviewers. I still would have liked to have seen a western blot of retinal homogenates that clearly demonstrates the presence of NCKX4 protein as recognized by the antibody used. NCKX proteins show a quite specific pattern of labeling, for example as shown in the Bronckers et al. paper that uses the validated Neuromab NCKX4 antibody, used here in the new immunohistochemistry images shown in Figure 2. Based on on the NCKX4 staining in the retina I would expect to see clear NCKX4 bands in a western blot of retinal homogenates of WT retinas (even more so in a western blot of retinal homogenates of the Nrl-/- mouse).

*Reviewer #3:*

The manuscript is improved.

---

## [Author Response]

*All three reviewers felt the paper made an important contribution to our understanding of cone function. Several points emerged as particularly important in the discussion among the reviewers:*

*1) Antibody validation. The immunohistochemistry is based on a newly developed antibody, and thus it is particularly important to validate its specificity. There are specific suggestions in the reviews, and the Journal of Comparative Neurology standards for reporting immunohistochemistry results also provide good guidelines. It is particularly important to include western blots of WT, NCKX KO and double KO.*

We have addressed this point in detail in our reply to the comment of reviewer 2 below. Briefly, we believe that our results already provide strong immunocutochemistry evidence for the specificity of our antibody. For example, Figure 2 clearly demonstrates that the conditional knockout of NCKX4 in cones results in selective loss of the NCKX4 antibody staining in these cells without affecting the staining in the inner retina. Western blot analysis of retinal homogenates will not provide any meaningful information in this case, because of the remaining NCKX4 expression in the retina and because cones are a very minor cell population in the retina. We have tried cone isolation but currently we cannot obtain sufficient quantities for western blots.

In an effort to address this point further, we performed immunocytochemistry experiments with a different NCKX4 antibody that was just validated by another group (Bronckers et al., 2017). The results we obtained were comparable to the ones with our own antibody and are now provided as an additional column in Figure 2 and mentioned in the corresponding Results section (subsection “The olfactory Ca^2+^ exchanger NCKX4 is expressed in the outer segments of mouse cones”, second paragraph).

*2) Rising phase analysis. The lack of effect of NCKX4 deletion on the rising phase of the light response is a key point of interpretation. This needs to be shown more clearly (i.e. on an expanded time base) and quantified.*

We have performed the requested analysis and now include the quantification of the rising phase of the cone response in control and NCKX4-deficient cones in the Results section (subsection “NCKX4 is required for the fast inactivation of mouse cone phototransduction”, last paragraph). We have revised Figure 4 to show in an inset the rising phase of the response on an expanded time scale as requested.

*3) Writing. More care is needed to distinguish the direct observations in the paper (e.g. impact of NCKX deletion on light responses) from inferences (e.g. impact on calcium levels). This is particularly an issue in the discussion.*

We have revised the manuscript, with particular attention to the Discussion to address this issue.

*Reviewer #1:*

*This paper explores the role of NCKX4 in cone photoreceptor function. The main conclusion is that NCKX2 and NCKX4 together regulate cone calcium levels and aspect of phototransduction that depend on calcium. The experiments generally appear well done and the data is clear. I have some concerns about the directness of the conclusions. Most importantly, in my view the paper could be considerably strengthened if a few of the interesting issues raised were explored in more depth.*

1) Amplification phase of light response.

*An important result in the paper is that NCKX4 deletion does not alter the rising phase of the light response. This is based in Figure 4. But the time base in the figure is too long to compare rising phases with any confidence, and quantification is needed to support the statement that the rising phases are unchanged as stated in the text.*

We have revised Figure 4 to include an inset showing the rising phase of the photoresponse on an expanded time scale. We have also included analysis of the leading edge of the photoresponse by determining the amplification constant of the rising phase in control and NCKX4-deficient cone responses: “analysis of the amplification constant in control and NCKX4-deficient cones revealed comparable (p>0.05) values of 0.5 ± 0.1 s^-2^ (N = 6) and 0.8 ± 0.3 s^-2^ (N = 6), respectively. […] Thus, the efficiency of phototransduction activation was not compromised by deletion of NCKX4 but light responses were slower, particularly in their recovery, in the Nckx4^f/f^ Cre^+^ cones compared to control mice, resulting in larger responses (Figure 4) and increased sensitivity of cones lacking NCKX4 (Figure 4; Table 1).”

2) Remaining adaptation.

*Adaptation appears to persist in the absence of both NCKX2 and NCKX4. This is presented as a surprise, but then is not pursued. The conclusion reached in the paper is that cones possess another means of regulating calcium; this rests on the assumption that there are not calcium-independent forms of adaptation, and I think this assumption should be stated clearly.*

Our data does not allow us to distinguish between calcium-dependent and calcium-independent mechanisms of light adaptation in NCKX2/4-deficient cones. One possibility is that calcium is still somehow extruded from cone outer segments even in the absence of both NCKX2 and NCKX4. Alternatively, it is possible that some residual calcium-independent adaptation is functioning in cones and is unmasked by our block of calcium extrusion. To address the reviewer’s request, we have explicitly stated the two possibilities in the revised Results (subsection “Cones require both NCKX2 and NCKX4 for normal structure and function”, last paragraph) and Discussion (subsection “Role of NCKX4 in cone phototransduction and light adaptation”, second paragraph).

3) Lack of NCKX2 data.

*The paper is weakened by the fact that the NCKX2 data was published elsewhere, and is not part of the present paper. One of the most interesting aspects of the paper is the apparent functional redundancy of the two proteins. The paper would be much stronger if complementary experiments were presented for cones lacking each protein individually and both together. Could some of the data from the previous paper be replotted here for comparison?*

To address the reviewer’s request, we have included dim flash responses from control and NCKX2-deficient cones as an inset to Figure 9 for comparison.

4) Lack of direct measures of calcium.

*In much of the paper, changes in response are equated with changes in calcium. In a few instances, the fact that this is an inference and not direct correspondence is clear, but in many cases the implications for changes in calcium are presented too strongly. I think it is quite important to separate the experimental observations from the inferences made about calcium signaling.*

We have revised the manuscript, with particular emphasis on the Discussion, in an effort to address this issue.

*Reviewer #2:*

*[…]*

*1) The immunohistochemistry is all based a novel NCKX4 antibody developed by the authors. I think for new antibodies it would be very helpful to carry out validation by demonstrating reactivity and specificity against in this case NCKX4 expressed in cell lines transfected with NCKX4 cDNA. The authors here show immunohistochemistry of WT and NCKX4 KO animals but only with cone-specific knockdown. Significant immunoreactivity is observed elsewhere in the retina. Have the authors carried out characterization of their antibody in cell lines expressing transfected NCKX4 or have they carried out immunohistochemistry in the NCKX2-4 DKO? Western blots of retinal extracts from WT and the various KO animals would address this well.*

We did not perform western blot analysis because the bulk of NCKX4 in the retina is not expressed in cones, but rather in rod bipolar cells as we show in Figure 1-3. Considering that cones represent only a small fraction of the neurons in the mouse retina (3% of all photoreceptors), and based on our experience with other cone-specific knockouts, we would expect that the reduction in the overall retinal NCKX4 expression in cone-specific NCKX4 knockout mice will be difficult to discern by western blots of retinal homogenates. We would respectfully reason that this issue is already clearly resolved by our immunocytochemistry results in Figure 2, which show the selective removal of NCKX4 immunoreactivity from cones but nor rod bipolar cells in cone-specific NCKX4 knockout mice. This result clearly demonstrates that our NCKX4 antibody is selective for NCKX4 and does not cross-react with NCKX2 or any other protein in cone photoreceptors. Essentially, this represents the experiment suggested by reviewer 2 but carried out in the relevant cone cells in the intact retina rather than in a cell line.

We agree with the reviewer that as we do not have a knockout of NCKX4 that blocks its expression in the rest of the retina, we cannot experimentally test the specificity of the NCKX4 antibody in rod bipolar cells. To address this issue, we have revised the corresponding section of the Results to now read: “These results suggest that in addition to its presence in the outer segments of mouse cone photoreceptors, NCKX4 might be expressed in rod bipolar cells, where it could potentially be involved in regulating retinal synaptic transmission and signal processing. However, further experiments will be required to confirm the expression of NCKX4 in rod bipolar cells and to determine its potential role for rod signaling.”

While we were working on the revision of our manuscript, a paper was published that used a commercial NCKX4 antibody which they validated in their studies (Bronckers et al., 2017, Calcif Tissue Int, 100: 80-86). To address the reviewer’s concern, we acquired and tested that antibody and obtained results comparable to the ones shown in the original submission of our manuscript with the use of our antibody. We have now included that data in Figure 2 and comment on the antibody validation more explicitly in the Results section (subsection “The olfactory Ca^2+^ exchanger NCKX4 is expressed in the outer segments of mouse cones”, second paragraph).

*2) I noticed that in Figure 2 the PNA and NCKX4 do not really co-localize but are found in different parts of the cone outer segment? This is in contrast to the zebrafish and macaque retinas where co-localization appears to be observed. Can the authors comment on this discrepancy?*

PNA labels the inner segment more intensely than the outer segment sheath in cones of many species that have been examined (for example, see “Specific binding of peanut lectin to a class of retinal photoreceptor cells. A species comparison” by Blanks and Johnson, IOVS 25:546-557). In addition, PNA staining pattern can be altered by tissue fixation differences. We think that the reviewer’s observation of a difference in localization is likely due to differences in fixation between the different tissues, leading to some alteration of PNA staining. In contrast, NCKX4 labeling is consistently observed in the outer segment in all species presented here.

*3) In the subsection “Cell-specificity of NCKX expression in the mouse retina”, the authors argue that lack of reduced dark current in either the NCKX2 or NCKX4 KO would be consistent with a lack of interaction between CNGA3 and either NCKX2 or NCKX4. Not sure if I am convinced about this argument as it would require careful measurements of NCKX2 and NCKX4 expression and although Matveev et al. failed to detect an interaction between NCKX2 and CNGA3, co-expression of both in HEK293 cells was shown to result in a direct interaction between these two proteins (Kang et al., Biochemistry 42: 4593).*

We agree with the reviewer that our results provide suggestive but certainly not definitive support for the lack of interaction between NCKX2/4 and the cone CNG channel. To address this point, we have revised the relevant text to now read: “The simplest explanation for this result is that, in contrast to the case in rods, the expression of NCKX does not control the expression or conductance of cone CNG channels. […] However, coexpression of NCKX2 and CNGA3 in HEK293 cells was found to results in direct interaction between these two proteins (Kang et al. 2003), indicating that this issue will require further investigation.”

*4) I would prefer to see the olfactory deleted from the title. Since NCKX4 was cloned it has been clear that NCKX4 had a broader expression pattern compared with say NCKX1 and the authors list in the introduction a number of other tissues in which NCKX4 has been shown to play an important role.*

We have removed “olfactory” from the Title and Abstract as suggested.

[Editors' note: further revisions were requested prior to acceptance, as described below.]

*The paper is much improved. There is a request for western blots to help validate the new NCKX antibody used in the paper. Including such data – specifically testing whether the antibody recognizes a protein with the correct MW – would help both validate data in the current paper and help determine how useful the new NCKX4 antibody is more generally.*

We have performed the western blot analysis and now include that data as Figure 2. The antibody recognizes a protein of ~50 kD. The molecular weight is comparable to the results obtained in the recently published data from Bronckers et al. using a commercial antibody (N414/25, NeuroMab, UC Davis/NIH NeuroMab Facility) which was validated using western blot of enamel organ from wildtype and NCKX4 knockout mice. As discussed in the letter accompanying our previous resubmission, we did not perform western blot analysis from the cone-specific NCKX4 knockout mice as the overall expression of NCKX4 in the retina is unlikely to change significantly due to the small number of cones. Further, we also purchased the N414/25 antibody and reproduced identical ICC images as those obtained using our antibody (see Figure 3). Following the successful validation of our NCKX4 antibody, we also removed the 2 panels in Figure 2 showing control immunostainings with the antibody recently used by Bronckers and colleagues.

*Reviewer #2:*

*I believe the authors have done an excellent job to address this reviewer's comments, and, as far as I can tell, also the comments by the other two reviewers. I still would have liked to have seen a Western blot of retinal homogenates that clearly demonstrates the presence of NCKX4 protein as recognized by the antibody used. NCKX proteins show a quite specific pattern of labeling, for example as shown in the Bronckers et al. paper that uses the validated Neuromab NCKX4 antibody, used here in the new immunohistochemistry images shown in Figure 2. Based on on the NCKX4 staining in the retina I would expect to see clear NCKX4 bands in a Western blot of retinal homogenates of WT retinas (even more so in a western blot of retinal homogenates of the Nrl-/- mouse).*

Done. Please see our reply to the Editorial Comments above.